# PTNet: A Proposal-centric Transformer Network for 3D Object Detection

**Jianping Zhong**[1]    **Zhaobo Qi**[1*]    **Kaiwen Duan**[2]    **Xinyan Liu**[1]
**Beichen Zhang**[1]    **Weigang Zhang**[1,3*]    **Qingming Huang**[2]
[1]Harbin Institute of Technology, Weihai
[2]University of Chinese Academy of Sciences
[3]Harbin Institute of Technology (Weihai) Qingdao Research Institute
`jianpingzhong@stu.hit.edu.cn, kaiwenduan@outlook.com,`
`{qizb, xinyliu, beiczhang, wgzhang}@hit.edu.cn, qmhuang@ucas.ac.cn`

## Abstract

3D object detection using LiDAR point cloud data is critical for autonomous driving systems. However, recent two-stage detectors still struggle to deliver satisfactory performance primarily due to inadequate proposal quality, which stems from significant geometric detail degradation in generated proposal features caused by high sparsity and uneven distribution of point clouds, as well as a complete failure to exploit surrounding contextual cues during independent proposal refinement, losing complementary details from adjacent proposals. To this end, we propose a **P**roposal-centric **T**ransformer **N**etwork (PTN), which includes a Hierarchical Attentive Feature Alignment (HAFA) and a Collaborative Proposal Refinement Module (CPRM). More concretely, HAFA employs a dual-stream architecture to extract multi-granularity proposal representations, including coarse-grained multi-scale voxel features and fine-grained coordinate point features to enhance proposals' object geometric representation ability. CPRM first generates hybrid object queries for all objects and then establishes contextual-aware interactions through the 3D parameter-guided deformable attention mechanism to effectively aggregate spatial location cues and category-specific information across proposals that are spatially adjacent and semantically correlated. Extensive experiments on the large-scale Waymo and KITTI benchmarks demonstrate the superiority of PTN. The code is available at `https://github.com/ZhongJianPing1/ptnet.git`.

## 1 Introduction

3D object detection serves as a foundational task for environmental perception in autonomous driving, aiming at precise localization and classification of objects in 3D scenes. Recently, to achieve a trade-off between performance and efficiency, researchers have focused on the two-stage 3D object detector paradigm Deng et al. (2021); Shi et al. (2023): this paradigm first employs the region proposal network (RPN) to obtain proposals, then extracts proposal features via ROI pooling Deng et al. (2021), and ultimately produces detection outputs by refining these proposal features in the subsequent refinement stage. However, existing two-stage 3D object detectors are constrained by suboptimal proposal quality, which arises from two key underlying issues.

The primary concern revolves around the degradation of geometric detail in the generated proposal features. In particular, prevailing two-stage detectors typically progressively expand receptive fields through pooling operations to generate proposal features. For areas of objects with few points, the pooling operation tends to filter out high-frequency geometric features such as surface details and edge sharpness, resulting in blurred proposal boundaries and structural incompleteness. As shown in Figure 1, certain objects have only a few points or sparsely distributed points. After multi-pooling operations, the detailed information inherent in these points is lost, leading to inaccurate predictions (denoted by green boxes). Although recent research has explored the use of foreground-agnostic

---
*Corresponding Authors.

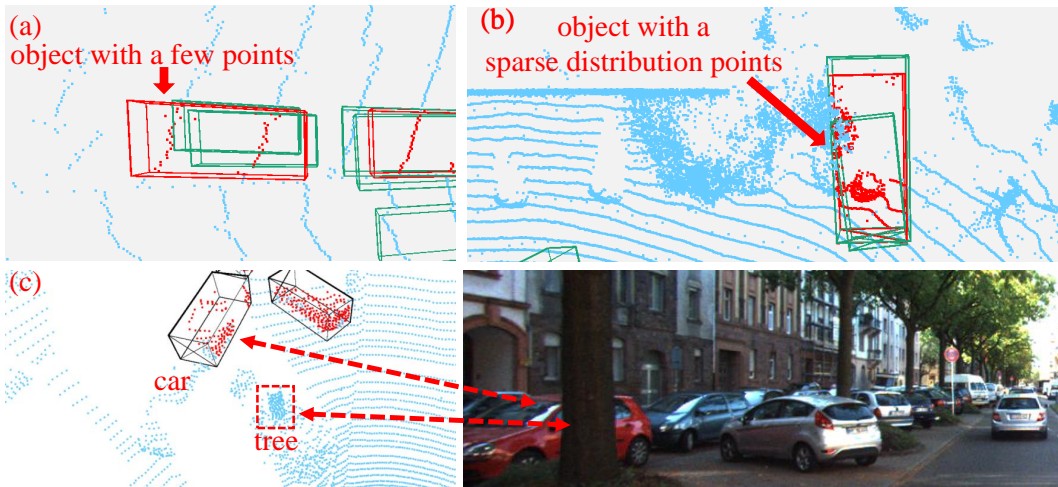

Figure 1: Qualitative results under the bird's-eye view (BEV) on Waymo. The red and black 3D boxes are annotated by humans, and the red points represent the object points. The green 3D boxes are predicted by VoxelNext. Some predicted boxes are not accurate due to corresponding objects having only a few points (a) or sparsely distributed points (b). (c) The car is occluded by the tree.

sampling methods to generate sampled raw points for compensating detailed information Shi et al. (2020; 2023), these methods tend to overlook sparse foreground points, which undermines the completeness of foreground representation and ultimately impairs the detection performance.

Another issue stems from the inefficient exploitation of surrounding contextual cues during the proposal refinement stage. Current two-stage methods optimize each proposal independently, only using its local features and failing to leverage complementary details from adjacent proposals with similar object characteristics. This limitation proves especially problematic in 3D scenes where occlusions are present. As illustrated in Figure 1 (c), since a car is occluded by a tree, its internal point clouds tend to be split into disjoint segments, with each segment independently predicted as a separate proposal. Due to the absence of cross-interaction with other proposals and the failure to integrate complementary information, the refined proposals suffer from inaccurate localization

To address the above challenges, we propose a Proposal-centric Transformer Network (PTN), which integrates a hierarchical attentive feature alignment module to enhance proposals' object geometric representation ability and a collaborative proposal refinement module to effectively aggregate complementary information among spatially adjacent and semantically correlated proposals.

The hierarchical attentive feature alignment module employs a dual-stream feature extraction architecture to capture complementary multi-granularity features to enhance proposal features. First, we propose a coarse-grained voxel feature extraction module to derive multi-scale semantic proposal features directly from voxel features, which enhances discriminative power for classification tasks. Concurrently, we design a fine-grained point feature retrieval module to recover intricate geometric details of proposals from unsampled raw foreground point clouds, thereby preserving precise spatial cues critical for regression refinement. Following the extraction of these dual-granularity features, we introduce a feature alignment module to harmonize them within a unified feature space, which ensures synergistic integration while maintaining their complementary strengths.

The collaborative proposal refinement module is designed to first generate object queries for all objects and then establish proposal contextual-aware interactions to extract complementary information from proposals that are spatially relevant and semantically similar. Specifically, we first select the top-$K$ proposals from the output of RPN based on their classification confidence scores to construct the basic proposal queries. Concurrently, we introduce learnable random queries to proactively explore objects that may have been overlooked, such as fully occluded or smaller objects. These hybrid queries are integrated as object queries. Next, we employ a 3D parameter-guided deformable attention mechanism to perform interaction between each object query and all generated proposals, which enables each object query to capture beneficial spatial location cues and category-specific semantic features from relevant proposals, thereby ultimately improving performance.

We conduct abundant experiments on large-scale 3D object detection benchmarks Waymo Sun et al. (2020) and KITTI Geiger et al. (2012). Experimental results demonstrate the effectiveness of PTN. In summary, our contributions are as follows:

- We introduce a hierarchical attentive feature alignment module, producing high-quality proposals with complementary multi-grained features.

- We propose an effective collaborative proposal refinement module, which adaptively aggregates crucial surrounding regions and performs proposal-level interaction.

- PTN achieves promising performance for 3D object detection on the large-scale datasets.

## 2 RELATED WORK

### 2.1 LiDAR-BASED 3D OBJECT DETECTION

Existing point cloud-based 3D object detection methods Fan et al. (2022b; 2024) predominantly fall into two technical paradigms: point-based and voxel-based approaches Zhong et al. (2025a;b). Point-based methods Chen et al. (2022) typically employ architectures like PointNet++ Qi et al. (2017) to directly extract features from unordered point clouds, followed by single-stage or two-stage detection frameworks to generate predictions. Given the massive scale of raw point clouds, such methods commonly adopt metric space sampling strategies to select representative point subsets for computational efficiency. Although point-based methods demonstrate superior performance on small datasets, their computational complexity scales linearly with point cloud cardinality. To address the efficiency bottleneck of point-based methods, researchers predominantly adopt voxel-based methods Liu et al. (2024); Jin et al. (2025); Zhang et al. (2024); Li et al. (2021); Fan et al. (2022b) to balance computational cost and performance. VoxelNet Zhou & Tuzel (2018) pioneers the transformation of irregular point clouds into structured voxel grids. Voxel R-CNN Deng et al. (2021) optimizes the two-stage detection pipeline, substantially reducing computational costs while maintaining accuracy. Our PTN is similar to Voxel R-CNN but introduces a core innovation: continuously refining the geometric quality and semantic consistency of proposals through hierarchical feature alignment and dynamic receptive field adjustment.

### 2.2 3D OBJECT DETECTION WITH DETR

Recent advancements in Transformer-based architectures have motivated extensive exploration of DETR Carion et al. (2020) frameworks for 3D point cloud object detection, particularly focusing on two critical design aspects: query initialization strategies and feature aggregation mechanisms. TransFusion Bai et al. (2022) leverages heatmap-guided localization to identify BEV feature peaks as initial queries, while CMT Yan et al. (2023) implements geometrically anchored learnable queries combined with global cross-attention for feature integration. Alternative solutions address specific limitations through innovative mechanisms. ConQueR Zhu et al. (2023) introduces contrastive query refinement to suppress false detections, and FocalFormer3D Chen et al. (2023) employs multiphase heatmap filtering alongside adaptive attention mechanisms to enhance both query selection efficiency and context modeling. However, existing DETR-inspired approaches still underperform compared to some non-transformer detectors. Compared to existing DETR-based approaches that rely on dense feature matching, we explicitly treat object proposals as learnable queries in the DETR framework. By enabling dynamic interaction between proposals through deformable attention mechanisms, we achieve more effective feature representation and information exchange.

## 3 METHODOLOGY

### 3.1 OVERVIEW

The framework of our PTN is shown in Figure 2. Given the point clouds $F_r$ as input, we first transform them into regular voxel representation. Next, we utilize a 3D backbone network Yan et al. (2018) to extract voxel features at three scales: $2\times$, $4\times$, and $8\times$ downsampled features denoted as $F_v^{N_v}$, where $N_v \in \{1, 2, 3\}$. Subsequently, we transform the $F_v^3$ features into the Bird's-Eye View (BEV) space and generate BEV features. A region proposal network (RPN) is then applied to produce proposals $B = \{b_i\}_{i=1}^N$ from the BEV features. Those proposals are also called region of interest (ROIs). For each proposal $b_i$, we employ the Hierarchical Attentive Feature Alignment (HAFA) module to enhance the proposal features, obtaining ROI features $f_b$. Afterward, we lever-

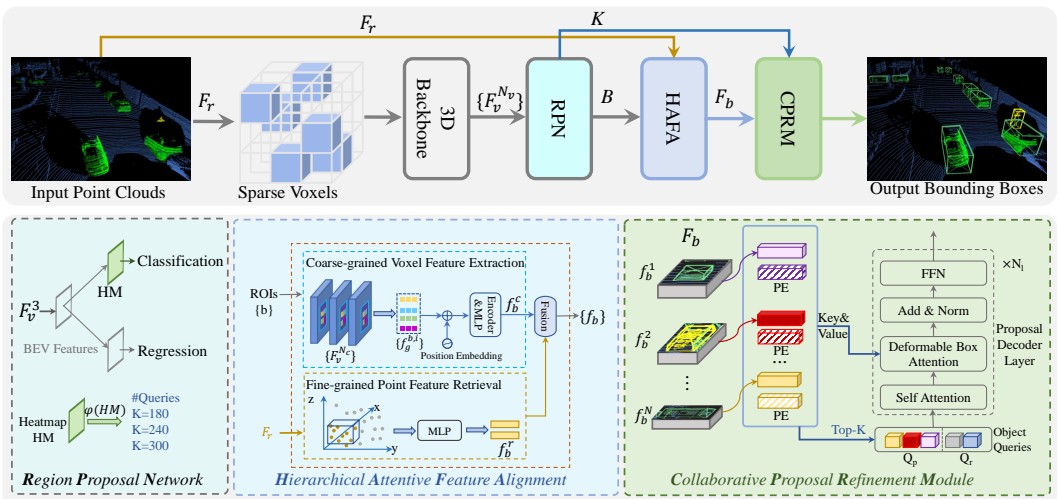

Figure 2: The overall architecture of PTN. It consists of a 3D backbone, an RPN, an HAFA module, and a CPRM module. Specifically, the HAFA uses a dual-stream feature extraction architecture to capture multi-granularity proposal features. The CPRM explicitly establishes contextual interactions among proposals through a hybrid query generation mechanism.

age the Collaborative Proposal Refinement Module (CPRM) to facilitate cross-proposal interaction among the proposals $B$. Finally, a feed-forward network (FFN) is used to predict the output.

## 3.2 HIERARCHICAL ATTENTIVE FEATURE ALIGNMENT

In this section, we propose a dual-stream feature extraction architecture to capture complementary multi-granularity features to enhance the proposal features. First, we propose a coarse-grained voxel feature extraction module to derive multi-scale proposal features from voxel features. Concurrently, we design a fine-grained point feature retrieval module to recover intricate geometric details from the unsampled raw point clouds. Following the dual-granularity feature extraction, we use a feature fusion module to harmonize them within a unified feature space.

### 3.2.1 COARSE-GRAINED VOXEL FEATURE EXTRACTION.

For each proposal $b$, we first use discrete grid points to represent it and then extract the corresponding grid point feature from the voxel feature by trilinear interpolation. Finally, we feed these grid point features into a Transformer-based encoder to enable cross-grid feature interaction within the proposal.

Given a proposal $b = (x, y, z, l, w, h, \theta)$, where $(x, y, z)$, $(l, w, h)$, and $\theta$ are the center, size, and rotation angle of the proposal. We uniformly divide the proposal into $g \times g \times g$ grid, and use the $g^3$ grid point $\mathcal{G}^b = \{(g_x, g_y, g_z)\}$ to represent the proposal $b$ following Voxel R-CNN Deng et al. (2021). After obtaining $\mathcal{G}^b$, we generate the grid points feature based on the voxel features $F_v^{N_v}$. Specially, we remap the grid points into voxel feature maps with different downsampling factors and apply trilinear interpolation to extract grid features $\{f_{g,N_v}^{b,i}\}_{i=1}^{g^3}$ and concatenate them to get multi-scale grid point features $\{f_g^{b,i}\}$. Then, we treat those multi-scale grid points as tokens and use them as the query content $Q_c = \{f_g^{b,i}\}_{i=1}^{g^3}$. Finally, we send the grid point features into the transformer encoder and MLP Layers to get the coarse-grained voxel feature $f_b^c$ as follows:

$$f_b^c = \text{MLP}(\text{Encoder}(Q_c, P_g)), \tag{1}$$

where $P_g$ is the position embedding. We employ a learned absolute position embedding function $\phi$ to encode the grid points position $p_g = \phi(g_x, g_y, g_z) \in \mathbb{R}^d$, $d$ is the channel dimension of $f_g^{b,i}$. For simplicity, we use $F_c = \{f_b^c\}$ to represent the coarse proposal features of $B$.

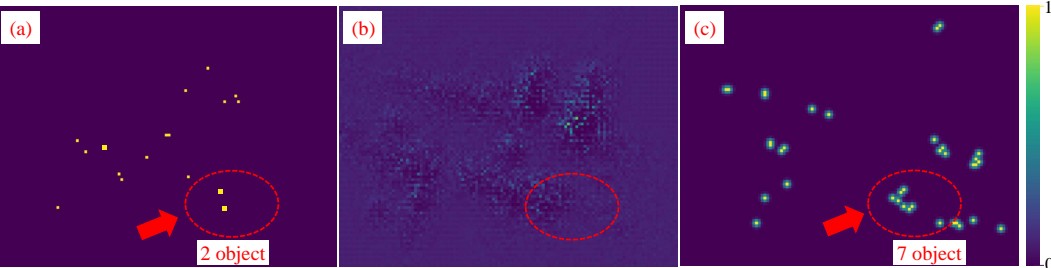

Figure 3: Three heatmaps for pedestrians. The left (a) displays a heatmap generated by applying local NMS to the predicted heatmap, while the middle (b) and right (c) show the predicted and ground-truth heatmaps, respectively. Notably, after applying local NMS, the red-circled area in the left heatmap filters out many correct predictions compared to the right heatmap.

### 3.2.2 FINE-GRAINED POINT FEATURE RETRIEVAL.

Given a proposal $b$ and the point clouds $P = \{p_x, p_y, p_z, f_a\}$, where $f_a \in \mathbb{R}^{C_p}$ are the intensity and timestamp features. we first select the foreground points $P' = \{(p_x, p_y, p_z)\}$ whose locations are inside $b$. Then we encode the geometric information of proposal $b$ into these foreground points to eliminate size ambiguity Li et al. (2021). Finally, we fuse those enhanced foreground points as the fine-grained point feature.

Specifically, for each proposal $b$, we translate its foreground point clouds into the coordinate system of the proposal and rotate them along the proposal direction angle $\theta_b$ through $P^* = R_\theta \cdot (P' - T_b)$, where $R_\theta$ denotes the rotation matrix and $T_b = (x, y, z)$ denotes the center of the proposal $b$. After obtaining the transformed foreground point clouds, we calculate the Euclidean distances from the transformed foreground points $P^*$ to the six surfaces of the proposal bounding box where they are located. The feature vector for each transformed foreground point is constructed as:

$$f_b^p = \text{Concat}((p_x^*, p_y^*, p_z^*), f_a, (d_l, d_r, d_f, d_b, d_t, d_d)), \tag{2}$$

where $d_{(.)}$ represents the Euclidean distance metric. These features are then processed by multi-layer perceptrons (MLPs) and max-pooling layers to generate fine-grained point features $f_b^r$:

$$f_b^r = \text{Maxpool}(\text{MLP}(f_b^p)), F_r = \{f_b^r\} \tag{3}$$

For simplicity, we use $F_r = \{f_b^r\}$ to represent the coarse proposal features of $B$.

### 3.2.3 FEATURE ALIGNMENT.

For multi-granular features, we first concatenate them into a composite feature representation and subsequently employ a convolutional network to project them into a unified feature space. The final proposal features $f_b$ are formulated as follows:

$$f_b = \text{Conv}(\text{Concat}(f_b^c, f_b^r)), F_b = \{f_b\} \tag{4}$$

## 3.3 COLLABORATIVE PROPOSAL REFINEMENT MODULE

In this section, we propose CPRM to generate high-quality proposal queries and random queries as object queries and then establish proposal contextual interactions with all proposals as complete contextual knowledge to achieve context awareness. Specifically, we first suppress spatial redundancies in RPN proposals $B$ via NMS to obtain candidate proposals $B_{nms}$. Then we dynamically select the top $K$ classification confidence proposals from $B_{nms}$ as high-quality proposal queries $Q_p$. Furthermore, we introduce $M$ learnable random queries $Q_r$ that serve as potential objects. These hybrid queries are integrated as object queries $Q = [Q_p, Q_r]$. Finally, a deformable cross-attention is applied to make interactions between hybrid object queries and the complete contextual knowledge $F_b$ that is preserved in $B$.

### 3.3.1 PROPOSAL QUERY GENERATION.

We first generate the candidate proposals $B_{nms}$ using the NMS. Generally, the proposals $B$ generated by RPN often exhibit high overlap ratios. The conventional methods address the overlap

issue by using Non-Local NMS Yin et al. (2021a). They utilize the centers of proposals for post-processing filtering, as shown in Figure 3. However, the classification performance is not very good in the early training; such methods tend to filter out true positive proposals that are close together (the red dashed circle area in Figure 3). To address this issue, we use the box of proposals for post-processing filtering. In particular, we employ box NMS with a low NMS threshold (e.g., 0.5) to select highly diverse candidate proposals $B_{nms}$. The corresponding features are $F_{nms}$.

Then we select the top $K$ proposals from $B_{nms}$ as proposal queries $Q_p$. As the object counts in 3D scenes are inherently uncertain, it is suboptimal to select fixed proposal queries. To achieve dynamic selection, we propose a query number estimation mechanism based on the classification score heatmap $HM$ in RPN.

Specifically, we first model the probability distribution of object counts $CNT$ in the scene as,

$$CNT = \begin{cases} \varphi(HM), & \text{if } epoch > \tau, \\ C, & \text{otherwise,} \end{cases} \tag{5}$$

where $\varphi(HM) = \text{SUM}(HM > st)$ is the function to calculate the number of objects. $st = 0.3$ is the classification score confidence to determine whether a proposal is an object. $\tau$ is the epoch to apply $HM$ to estimate object counts. $C$ is the maximum object count.

Then we categorize $CNT$ into three intervals $\{[n_k^{\min}, n_k^{\max})\}_{k=1}^3$ and adaptively set the proposal queries number $K$ according to:

$$K = \begin{cases} K_1, & \text{if } n_1^{\min} \leq CNT < n_1^{\max} \\ K_2, & \text{if } n_2^{\min} \leq CNT < n_2^{\max} \\ K_3, & \text{if } n_3^{\min} \leq CNT < n_3^{\max} \end{cases} \tag{6}$$

Finally, we select the proposal based on the classification confidence of the $B_{nms}$ and $K$ as follows:

$$Q_p = \text{top}_K(F_{nms}, HM), P_p = \phi(\text{top}_K(B_{nms}, HM)) \tag{7}$$

where $\text{top}_K(u, v)$ means select top $K$ queries from $u$ according to $v$. $P_p$ is the position embedding. $K$ is determined by Equation 6. $HM$ is the classification score heatmap in RPN.

### 3.3.2 RANDOM QUERY GENERATION.

After getting the high-quality object queries, we preserve random queries $B_r$ to retrieve some over-looked objects by $Q_p$. In particular, the BEV space is uniformly partitioned into $X \times Y$ grids (the BEV feature map size is $X \times Y \times C$), where each grid center initializes an auxiliary query. The centers of random queries $B_r$ align with grid positions, and their scales are uniformly set to $(0.05L, 0.05W, 0.5H)$. By incorporating such spatially prior-constrained random queries, our method effectively recalls TPs over-suppressed by NMS while maintaining high detection precision. The random queries content $Q_r$ and position embedding $P_r$ are defined as follows:

$$Q_r \in \mathbb{R}^d, P_r = \phi(B_r) \tag{8}$$

where $d$ is the channel dimension of the $f_b$.

The object queries are donated as $Q = \text{Concat}(Q_p, Q_r)$, and the position embedding $P_q = \text{Concat}(P_p, P_r)$.

### 3.3.3 PROPOSAL TRANSFORMER DECODER.

This module aims to establish an interaction between the generated object queries and the complete contextual knowledge $B$ to achieve object-level context awareness. A direct method is to follow the DETR by using cross-attention. However, such a paradigm neglects the inherent differences between 2D images and 3D point clouds. Unlike 2D images where objects may occupy most of the image, 3D objects exhibit spatial sparsity and occupy minimal area. Consequently, 3D proposal interactions should focus exclusively on neighborhoods, which reduces computational redundancy while enhancing geometric relationship modeling. Specifically, for each object query, we utilize its 3D bounding box parameters (e.g., position, dimensions, orientation) to generate spatial attention

Table 1: Comparison with prior methods on the Waymo Open dataset (single-frame setting). Metrics: mAP/mAPH (%)↑ for the overall results, and AP/APH (%)↑ for each category. ‡: two-stage method. †: detr-like methods. −: results are not published.

| | | | | | | | |
|---|---|---|---|---|---|---|---|
| *Results on the validation dataset* | | | | | | | |
| Methods | mAP/mAPH L2 | Vehicle AP/APH L1 | L2 | Pedestrian AP/APH L1 | L2 | Cyclist AP/APH L1 | L2 |
| CenterPoint Yin et al. (2021b) | 68.2/65.8 | 74.2/73.6 | 66.2/65.7 | 76.6/70.5 | 68.8/63.2 | 72.3/71.1 | 69.7/68.5 |
| PV-RCNN‡ Shi et al. (2020) | 69.6/67.2 | 78.0/77.5 | 69.4/69.0 | 79.2/73.0 | 70.4/64.7 | 71.5/70.3 | 69.0/67.8 |
| SST_TS‡ Fan et al. (2022a) | –/– | 76.2/75.8 | 68.0/67.6 | 81.4/74.0 | 72.8/65.9 | –/– | –/– |
| SWFormer† Sun et al. (2022) | –/– | 77.8/77.3 | 69.2/68.8 | 80.9/72.7 | 72.5/64.9 | –/– | –/– |
| PillarNet-34 Shi et al. (2022) | 70.9/68.4 | 79.1/78.6 | **70.9/70.5** | 80.6/74.0 | 72.3/66.2 | 72.3/71.2 | 69.7/68.7 |
| CenterFormer†Zhou et al. (2022) | 71.1/68.9 | 75.0/74.4 | 69.9/69.4 | 78.6/73.0 | 73.6/68.3 | 72.3/71.3 | 69.8/68.8 |
| PV-RCNN++‡ Shi et al. (2023) | 71.0/64.9 | 78.8/78.2 | 70.3/69.7 | 76.7/76.2 | 68.5/59.7 | 69.0/67.6 | 66.5/65.2 |
| TransFusion† Bai et al. (2022) | –/64.9 | –/– | –/65.1 | –/– | –/63.7 | –/– | –/65.9 |
| DSVT Wang et al. (2023) | 73.2/71.0 | **79.3/78.8** | **70.9/70.5** | 82.8/77.0 | 75.2/69.8 | 76.4/75.4 | 73.6/72.7 |
| ConQueR† Zhu et al. (2023) | 70.3/67.7 | 76.1/75.6 | 68.7/68.2 | 79.0/72.3 | 70.9/64.7 | 73.9/72.5 | 71.4/70.1 |
| FlatFormer† Liu et al. (2023) | 69.7/67.1 | –/– | 69.0/68.6 | – / – | 71.5/65.3 | – / – | 68.6/67.5 |
| Shift-SSD Chen et al. (2024) | 64.8/61.1 | 74.1/73.6 | 65.1/64.6 | 72.3/ 62.3 | 63.4/ 54.5 | 68.2 /66.4 | 66.0/ 64.2 |
| LiDAR-PTQ Zhou et al. (2024) | 67.6/65.1 | –/– | 66.2/65.7 | –/– | 67.9/62.2 | –/– | 68.6/67.5 |
| DRET Huang et al. (2024) | 71.0/68.6 | 78.5/78.0 | 70.0/69.5 | 81.0/75.1 | 72.2/66.7 | 73.4/72.5 | 70.7/69.7 |
| PASS-PV Chen et al. (2025) | 72.0/65.7 | 78.3/**78.8** | 70.5/70.0 | 76.2/66.9 | 67.2/58.8 | 71.8/70.7 | 69.4/68.3 |
| PTN | **73.5/71.2** | 76.7/77.1 | 68.7/68.2 | **84.2/78.6** | **76.8/71.4** | **77.7/76.5** | **75.0/73.9** |
| *Results on the testing dataset* | | | | | | | |
| Methods | mAP/mAPH L2 | Vehicle AP/APH L1 | L2 | Pedestrian AP/APH L1 | L2 | Cyclist AP/APH L1 | L2 |
| PV-RCNN++ Shi et al. (2023) | 72.4/70.2 | **81.6/81.2** | **73.9/73.5** | 80.4/75.0 | 74.1/69.0 | 71.9/70.8 | 69.3/68.2 |
| PillarNet Shi et al. (2022) | 70.1/67.1 | 81.1/80.6 | 73.6/73.2 | 78.3/70.2 | 72.2/64.6 | 67.2/66.0 | 64.7/63.6 |
| PillarNeXt Li et al. (2023) | 72.2/69.6 | –/– | –/– | –/– | –/– | –/– | –/– |
| Fade3D Ye et al. (2025) | –/– | 77.7/77.2 | 69.9/69.5 | –/– | –/– | –/– | –/– |
| PTN | **72.7/70.6** | 80.2/79.8 | 72.5/72.1 | **82.0/77.0** | **76.0/71.2** | **72.2/71.1** | **69.6/68.5** |

weights. These weights dynamically adjust the sampling offset of deformable convolution kernels. This enables the network to autonomously capture the structural features of neighboring objects and to enrich each object query with scene-level dependencies. During proposal refinement, the detector jointly optimizes each object query by leveraging these scene-level dependencies. The enhanced object queries $Q' = \text{Decoder}(Q, P_q, F_b)$ are used for the subsequent detection head.

# 4 EXPERIMENTS

## 4.1 DATASETS AND EVALUATION METRICS

*Waymo Open Dataset (Waymo).* It includes 798, 202, and 150 scenes for the training, validation, and testing sets. It provides three categories: vehicle, pedestrian, and cyclist. The evaluation uses mean average precision (mAP) and mAP weighted by heading accuracy (mAPH). Objects are classified into two levels: LEVEL 1 (L1) for more than 5 point clouds and LEVEL 2 (L2) for more than 1.

*KITTI.* There are 7481 and 7518 samples for training and testing. The dataset includes three categories: car, pedestrian, and cyclist. The 7481 training samples are divided into two parts: 3769 and 3712 samples for the training and validation sets. We use 3D mAP as the evaluation metric.

## 4.2 IMPLEMENTATION DETAILS

For Waymo and KITTI, we apply PTN to Voxel R-CNN. The setting aligns with prior works Deng et al. (2021). In CPRM, we categorize CNT into three intervals $[0, 20)$, $[20, 40)$, and $[40, 200]$, and set the parameters as $K_1 = 180, K_2 = 240, K_3 = 300$. Additional results are provided in the supplementary materials.

Table 3: Effectiveness of PTN on Waymo validation set using multi-frame inputs.

| Methods | Frames | mAP/mAPH (L2) |
|---|---|---|
| CenterPoint | 4 | 70.8/69.4 |
| CenterFormer Zhou et al. (2022) | 4 | 74.7/73.2 |
| PillarNet Shi et al. (2022) | 2 | 72.2/68.4 |
| PTN | 3 | 74.5/73.2 |
| PTN | 4 | **75.6/74.1** |

Table 2: Comparison on KITTI. Metrics: mAP↑ for the overall results. −: results are not published.

| | *Results on the testing data set* | | | | | | | |
|---|---|---|---|---|---|---|---|---|
| Methods | Car | | | | Cyclist | | | |
| | Easy | Moderate | Hard | mAP | Easy | Moderate | Hard | mAP |
| HVPRNoh et al. (2021) | 86.38 | 77.92 | 73.04 | 79.11 | – | – | – | – |
| SASA Chen et al. (2022) | 88.76 | 82.16 | 77.16 | 82.69 | – | – | – | – |
| IA-SSDZhang et al. (2022) | 88.34 | 80.13 | 75.04 | 81.44 | 78.35 | 61.94 | 55.70 | 68.30 |
| Voxel R-CNN Deng et al. (2021) | 88.09 | 80.99 | 76.50 | 81.86 | 76.42 | 62.01 | 55.94 | 64.79 |
| PASS-PV Chen et al. (2025) | 87.65 | 81.28 | 76.79 | 81.90 | 80.43 | 68.45 | 60.93 | 69.93 |
| DPFusion Mo et al. (2025) | 90.98 | 82.35 | 77.26 | 83.53 | 79.96 | 66.47 | 58.47 | 68.30 |
| PTN | **91.60** | **82.77** | **77.96** | **84.11** | **83.38** | **70.30** | **62.63** | **72.11** |
| | *Results on the validation data set* | | | | | | | |
| Methods | Car | | | | Cyclist | | | |
| | Easy | Moderate | Hard | mAP | Easy | Moderate | Hard | mAP |
| EPNetHuang et al. (2020) | 88.76 | 78.65 | 78.32 | 81.91 | 83.88 | 65.60 | 62.70 | 70.72 |
| Pointformer Pan et al. (2021) | 87.13 | 77.06 | 69.25 | 77.81 | 75.01 | 59.80 | 53.99 | 62.93 |
| IA-SSD Zhang et al. (2022) | 91.88 | 83.41 | 80.44 | 85.24 | 88.42 | 70.14 | 65.99 | 74.85 |
| VFF Li et al. (2022) | 92.31 | 85.51 | **82.92** | 86.91 | 89.40 | **73.12** | **69.86** | 77.46 |
| Voxel R-CNN Deng et al. (2021) | 92.53 | 85.03 | 82.56 | 86.70 | 89.52 | 72.62 | 68.32 | 76.82 |
| Fade3D Ye et al. (2025) | 90.92 | 82.00 | 77.49 | 83.47 | – | – | – | – |
| PTN | **92.74** | **85.92** | 82.87 | **87.17** | **91.87** | 72.66 | 68.32 | **77.61** |

## 4.3 STATE-OF-THE-ART COMPARISON

***Waymo.*** Results in Table 1 demonstrate that PTN surpasses most detectors. PTN shows significant improvement in pedestrian. On one hand, the point clouds of pedestrians typically exhibit sparse distributions, and the use of coarse-grained voxel features tends to cause localization inaccuracies. HAFA addresses this by incorporating fine-grained point cloud features to enhance precise localization. On the other hand, CPRM facilitates interaction between current objects and surrounding ones, effectively mitigating occlusion and overlapping issues. The results obtained by using multi-frame data as input in Table 3 demonstrate that PTN outperforms existing detection approaches.

We compare PTN with other methods in terms of performance and inference speed, as depicted in Figure 4. Remarkably, PTN achieves a good trade-off between performance and inference speed. All models are evaluated on the NVIDIA A100 GPU.

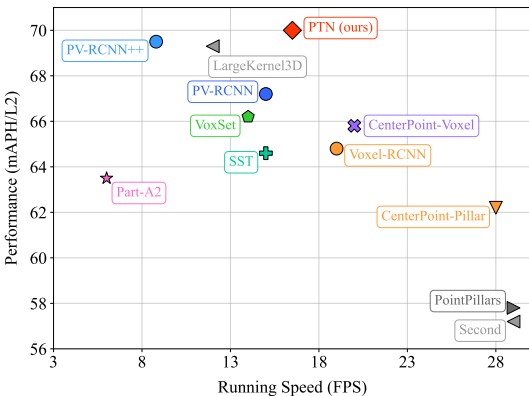

Figure 4: Performance and speed of our PTN and other leading performance detectors on Waymo.

***KITTI.*** The results on KITTI are shown in Table 4.1. PTN achieves promising results on the KITTI test set, particularly in detecting cyclists. This can be attributed to the fact that most of the cyclists' area is empty, making it difficult for the model to detect them with limited points. In contrast, PTN introduces fine-grained points to complete the appearance of these objects, making them easier to detect. PTN is generalizable and can be applied to most of the datasets.

## 4.4 ABLATION STUDIES

**The efficiency of each component.** In Table 4, we present an ablation study for different components on Waymo. To improve efficiency, we use 25% of the data for training and validation. In the following section, we use this as the default setting unless otherwise stated. CPRM explicitly introduces the relation between different proposals, which is beneficial for the detector to filter out background noise and improve preci-

Table 4: Ablation study of each component on Waymo.

| HAFA | CPRM | 3D AP/APH (L2) | | | mAP/mAPH (L2) |
|---|---|---|---|---|---|
| | | Vehicle | Pedestrian | Cyclist | |
| | | 59.09/58.57 | 58.20/52.47 | 62.73/61.33 | 60.00/57.45 |
| ✓ | | 60.86/60.38 | 65.13/57.13 | 64.58/63.08 | 63.52/60.19 |
| | ✓ | 61.91/61.49 | 65.62/58.69 | 65.82/64.54 | 64.45/61.57 |
| ✓ | ✓ | **62.92/62.50** | **69.23/62.37** | **67.05/65.88** | **66.40/63.58** |

Table 5: Ablation study for each component in PTN on Waymo. $N_p$ and $N_r$ represent the number of object queries and random queries, respectively.

| $N_p$ | $N_r$ | 3D AP/APH (L2) | | | mAP/mAPH (L2) |
|---|---|---|---|---|---|
| | | Vehicle | Pedestrian | Cyclist | |
| 200 | 0 | 54.57/54.01 | 62.81/53.04 | 60.05/58.47 | 59.14/55.17 |
| 300 | 0 | 58.35/57.83 | 64.96/55.34 | 61.94/60.21 | 61.75/57.79 |
| 400 | 0 | 55.59/55.03 | 62.88/53.53 | 60.89/59.28 | 59.78/55.94 |
| 300 | 100 | **61.91/61.49** | **65.62/58.69** | **65.82/64.54** | **64.45/61.57** |

sion. HAFA assists in locating the bounding box of objects, thereby improving the accuracy of the detector in determining the location and shape of objects.

**The number of object queries.** When the number of proposal queries increases (e.g., $N_q = 400$ vs $N_q = 300$), the similarity among queries rises significantly, leading to feature redundancy and consequent degradation in classification performance (from 57.79% mAPH to 55.94% mAPH) as shown in Table 5. Conversely, reducing the number of proposal queries (e.g., $N_q = 200$ vs $N_q = 300$), results in missed detection of low-score true positive (TP) samples, thereby lowering the recall (from 57.79% mAPH to 55.17% mAPH). This indicates a fundamental trade-off between the diversity and similarity of proposal queries. To mitigate this conflict, we introduce random queries, which enable the recovery of overlooked low-score TPs.

**The results in sparse scenarios.** In Table 6, we test the performance in sparse scenarios. We first construct sparse scenarios and then measure performance by counting the number of True Positives (TPs). Specifically, we sort ob-

Table 6: Number of true positives under different IoU in sparse scenarios on Waymo.

| IoU | 0.1 | 0.3 | 0.5 | 0.7 |
|---|---|---|---|---|
| Voxel R-CNN | 10327 | 7017 | 4372 | 1075 |
| PTN | 10656(+3.1%) | 7401(+5.4%) | 4712(+7.7%) | 1193(+10.9%) |

jects in descending order according to their internal point counts. We then use the last 2.5% of objects to construct sparse scenarios. We count the number of TPs on both the baseline and PTN at different IoU thresholds. The results demonstrate the efficiency of PTN in sparse scenarios. It is noteworthy that PTN exhibits significantly enhanced performance under high IoU thresholds. This improvement is attributed to the HAFA, which restores essential details aiding localization.

**The performance on occluded scenes.** We divide the scene into heavily occluded scenarios (the objects whose internal point cloud number is less than 20 or the distance from the LiDAR sensor to them exceeds 50 meters) and lightly occluded scenarios (other objects). Results in Table 7 indicate that PTN shows a relatively smaller decrease in recall compared to the Voxel R-CNN Deng et al. (2021). When the

Table 7: Recall@0.5 at different occluded scenes on Waymo. $^-$: no random queries. LO and HO represent lightly and heavily occluded, respectively. VR represents Voxel R-CNN.

| Setting | VR | PTN$^-$ | PTN | PTN$^-$ improv. | PTN improv. |
|---|---|---|---|---|---|
| LO | 0.825 | 0.856 | 0.874 | 3.1% | 4.9% |
| HO | 0.634 | 0.685 | 0.722 | 5.1% | 8.8% |

top $K$ proposals do not include these objects, it becomes challenging for the object queries to recall them effectively. For the objects missed in the proposals, we add random queries to the object queries to interact with complete contextual knowledge, thereby improving recall.

**The efficiency of the HAFA.** When only using the fine-grained point features to refine the proposals, the performance is better than using the voxel features, as shown in Table 8. This is because the classification results are encoded into the object query content, while the proposal locations are encoded into the object queries' position embedding.

Table 8: Ablation study of HAFA on Waymo. CVFE, FPFR, and FA represent coarse-grained voxel feature extraction, fine-grained point feature retrieval and feature alignment, respectively.

| CVFE | FPFR | FA | 3D AP/APH (L2) | | | mAP/mAPH (L2) |
|---|---|---|---|---|---|---|
| | | | Vehicle | Pedestrian | Cyclist | |
| ✓ | | | 59.09/58.57 | 58.20/52.47 | 62.73/61.33 | 60.00/57.45 |
| | ✓ | | 60.66/59.21 | 58.26/52.52 | 62.54/61.19 | 60.48/57.64 |
| ✓ | ✓ | | 58.91/58.37 | 56.73/50.35 | 61.74/60.12 | 59.12/56.28 |
| ✓ | ✓ | ✓ | **60.86/60.38** | **65.13/57.13** | **64.58/63.08** | **63.52/60.19** |

## 5 CONCLUSION

In this paper, we propose PTN, a novel Proposal-centric Transformer Network for 3D object detection. Since the performance of existing two-stage detectors is limited by the quality of proposals in terms of fine-grained information decay and the lack of effective exploitation of contextual cues, we address these issues with PTN. PTN aims to enhance the proposal features for accurate 3D detection. Specifically, we use a dual stream feature extraction module to extract coarse grained voxel features and fine-grained point features, and align them to enhance the representation of proposals.

Furthermore, we propose a collaborative proposal refinement module to explicitly establish contextual interactions among proposals through a proposal transformer decoder. Extensive experiments on the KITTI and Waymo benchmarks demonstrate the effectiveness of PTN. Future work will focus on improving proposal quality through learnable mechanisms with minimal cost.

REPRODUCIBILITY STATEMENT

To ensure the reproducibility of our work, we provide general details on the datasets and experimental settings in Section 4. Comprehensive information on the model architecture, datasets, and training strategies can be found in Appendix A.

## ACKNOWLEDGEMENT

This work is partially supported by the National Natural Science Foundation of China under Grants 62441232, 62476068, 62306092, and 62502115, and supported by Shandong Provincial Natural Science Foundation under Grants ZR2025ZD01, ZR2024QF066, and ZR2025QC1516.

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

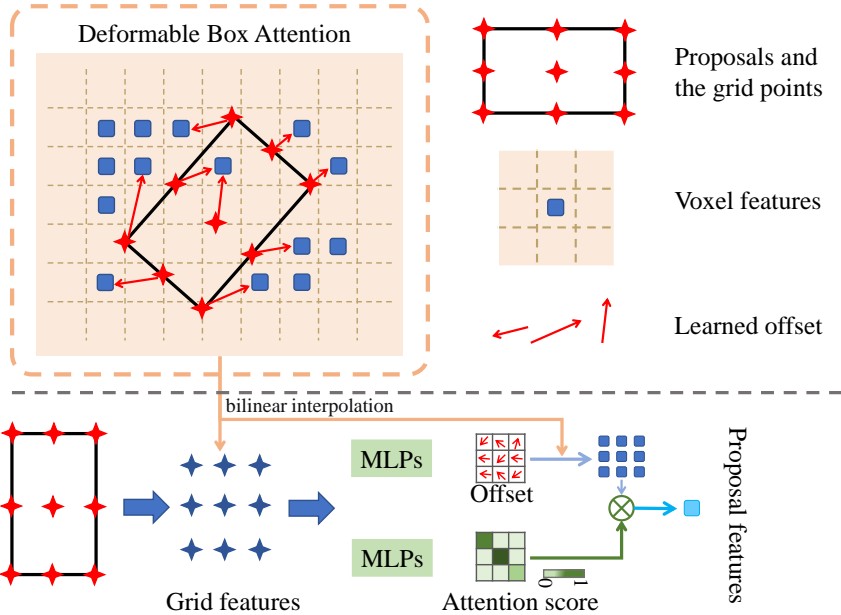

Figure 5: The detail of the deformable box attention.

APPENDIX

THE USAGE OF LLMS

In this work, large language models (LLMs) are employed solely during the manuscript preparation stage to assist with translation and language refinement. Beyond this purpose, they are not utilized for any other aspects of the study.

## A  DETAILS OF THE PTN

**Details of DBA.** The detailed structure of the Deformable Box Attention (DBA) module is illustrated in Figure 5. The core idea of DBA is to divide each proposal into $G^2$ grid points and learn an offset for every grid, enabling it to aggregate features from neighboring proposals. The grid features are then fused through learnable attention weights to obtain the final interaction-enhanced proposal representation. This grid-based decomposition provides two key benefits: (1) using multiple grid locations offers richer semantics and spatial information than relying solely on the proposal center; (2) the grid layout implicitly encodes proposal size, which strengthens both regression and classification. Specially, given a proposal $b_i = (x, y, z, l, h, w, \theta)$, where $(x, y, z)$ denotes the center coordinates, $(l, h, w)$ represents the dimensional size, and $\theta$ indicates the orientation, we first normalize its coordinates to range [0, 1] relative to point cloud range $(x_0, y_0, z_0, x_1, y_1, z_1)$ as,

$$b_i^* = \left( \frac{x - x_0}{x_1 - x_0}, \ \frac{y - y_0}{y_1 - y_0}, \ \frac{z - z_0}{z_1 - z_0}, \ \frac{l}{x_1 - x_0}, \right.$$
$$\left. \frac{h}{y_1 - y_0}, \ \frac{w}{z_1 - z_0}, \ \frac{\theta + \pi}{2\pi} \right). \tag{9}$$

Subsequently, we convert the proposal into a set of $G^2$ grid points $P_g$, as shown in Figure 5. These grid points are projected onto the proposal voxel feature map F to extract the corresponding grid features $F_g = \{f_g^k\}_{k=1}^{G^2}$. For all grids, we first compute the normalized offset $O_g = \sigma(MLP(F_g))$ from $F_g$, where $\sigma$ denotes the sigmoid function to normalize the offset into [0,1]. The final sampling locations are the normalized offset locations $O_g$ plus the initial grid locations Pg. Based on the sampling locations, we apply bilinear interpolation on the voxel feature map to obtain the sampled grid features $F_g = \{f_g^k\}_{k=1}^{G^2}$. The learned offsets allow grids to sample features from nearby proposals, naturally enabling cross-proposal feature interaction. In parallel, we compute attention weights for all grids: $A = softmax(MLP(F_g)) = \{a_k\}_{k=1}^{G^2}$. Finally, the enhanced proposal representation is obtained by a weighted sum over all sampled grid features: $F_p = \sum_{k=1}^{G^2} a_k \cdot f_s^k$. This design enables rich feature exchange between proposals and implicitly encodes geometric structure, leading to stronger proposal refinement.

## B    DETAILS OF THE EXPERIMENTAL SETTING

**The intervals setting of the ground truth.** In the CPRM, we categorize the estimated object number $CNT$ into three intervals $\{[n_k^{\min}, n_k^{\max}]\}_{k=1}^3$. In the experimental implementations, we set them as $[0, 20)$, $[20, 40)$, and $[40, 200]$. This setting is based on the object distribution of the dataset. As shown in Figure 6, we analyzed the distribution of object counts in the Waymo validation set and calculated the 30%, 60%, and 90% percentiles, which are roughly around 20, 40, and 100, respectively. Additionally, the maximum number of objects in a single scene in the dataset is reached 200. For other datasets, we can also set the intervals based on the object count distribution.

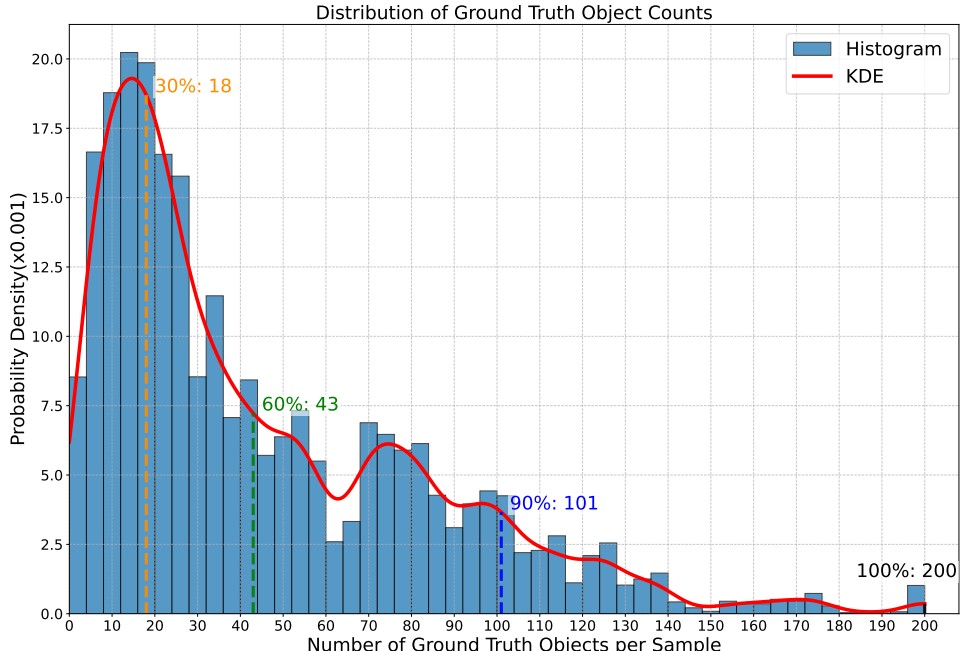

Figure 6: The object number distribution on the Waymo.

## C    MORE EXPERIMENT RESULTS

**The speed and cost of each component.** Table 9 details the performance and cost of the individual modules on Voxel-RCNN Deng et al. (2021). The results indicate that adding HAFA improves performance by 3.52/2.74 in mAP/mAPH

Table 9: Inference cost and speed of each component.

| HAFA | CPRM | mAPH (L2) | Speed (FPS) | Memory(G) | FLOPs(G) |
|------|------|-----------|-------------|-----------|----------|
|      |      | 57.45     | **19.5**    | 4533      | 101.8    |
| ✓    |      | 60.19     | 18.0        | 2431      | 21.7     |
|      | ✓    | 61.57     | 17.6        | 2027      | 63.4     |
| ✓    | ✓    | **63.58** | 16.5        | 2995      | 69.6     |

while reducing speed by 1.5 FPS. Adding CPRM improves performance by 3.35/4.12 mAP/mAPH and reduces speed by 1.9 FPS. When both modules are integrated, PTN yields the best performance of 66.40/63.58 mAP/mAPH with a speed of 16.5 FPS. PTN strikes a balance between performance and inference speed. Furthermore, PTN exhibits reduced computational complexity, with fewer FLOPs and lower memory usage compared to the baseline.

**The generality of PTN.** To validate the effectiveness of HAFA and CPRM, we applied them to PV-RCNN, with the results summarized in Table 10. The experimental results demonstrate that incorporating only HAFA yields significant improvements, particularly for pedestrians (ped) and cyclists (cyc). This is attributed to the sparse point cloud distribution of these

Table 10: Performance on PV-RCNN on Waymo.

| Methods | 3D AP/APH (L2) | | |
|---------|------|------|------|
|         | *Veh* | *Ped* | *Cyc* |
| PV-RCNN(PV) | 68.0/67.5 | 67.6/61.6 | 67.7/66.5 |
| PV+HAFA | 68.3/67.9 | 68.4/62.3 | 68.1/67.2 |
| PV+CPRM | 68.8/68.2 | 69.2/63.2 | 68.7/67.6 |
| PV+PTN | **69.1/68.6** | **70.3/64.4** | **69.0/67.9** |

objects, where edge details are prone to being lost during downsampling operations. HAFA effectively mitigates this by recovering these critical details. When CPRM is incorporated solely, performance gains are observed in all categories, with the most pronounced improvements in pedes-

trians (ped) and vehicles (vehicle). The abundance of training samples for these categories allows CPRM to facilitate enhanced learning of neighboring contextual information, thereby optimizing both classification and regression performance. The concurrent integration of both modules leads to optimal performance. This result convincingly demonstrates the generalizability of our approach.

**The threshold value of score** $st$**.** We preliminarily estimate the number of objects in scene through the score distribution of first-stage heatmaps, thereby providing prior knowledge for the query quantity of objects in CPRM.

Table 11: Mean average error on estimated object number with different score threshold on Waymo.

| $st$ | 0.1 | 0.2 | 0.3 | 0.5 | 0.7 |
|------|-----|-----|-----|------|------|
| MAE | 13 | 10 | 6 | $\lvert-10\rvert$ | $\lvert-17\rvert$ |

During this process, we first apply local Non-Maximum Suppression (NMS) to the heatmap to remove overlapping points. We then set a score threshold: regions with scores exceeding this threshold are considered target objects, while others are rejected. To obtain accurate object counts, we test different score thresholds (experimental results shown in Table 11). The results indicate that when the threshold is too low, the estimated quantity significantly exceeds the actual number; conversely, an excessively high threshold tends to miss targets. Experiments demonstrate that at a score threshold of 0.3, the gap between estimated and actual counts is minimized. Therefore, we set the score threshold to 0.3 in the experiments.

**The Process of Overlapping Proposals.** To achieve high recall, we do not handle overlapping boxes in HAFA, as those overlapping proposals serve as keys and values in the subsequent decoder. Instead, we address this issue in CPRM: before selecting the $topK$ proposals

Table 12: Performance of PTN using different IoU thresholds in the NMS on Waymo.

| IoU | 0.3 | 0.4 | 0.5 | 0.6 | 0.7 |
|-----|-----|-----|-----|-----|-----|
| mAPH | 60.68 | 62.84 | **63.58** | 61.41 | 59.96 |

as object queries, we apply NMS post-processing to suppress redundant regions, thereby alleviating conflicts during Hungarian matching. The results in Table 12 indicate a balance between overlap and differences. As the IoU increases, the overlap among retained proposals becomes greater while their differences decrease. This leads to more pronounced conflicts among the classification branches during Hungarian matching, ultimately limiting overall performance.

**The Robustness of PTN.** In DETR-like methods Liu et al. (2024); Carion et al. (2020); Zhu et al. (2021), the number of object queries represents the maximum number of objects that the detector can predict in a scene and is closely tied to the dataset's object distribution. Properly adjusting the number of object queries is crucial. Too many may complicate training and increase computational resource requirements, while too few may result in missing some ob-

Table 13: Ablation study for CPRM component in PTN on KITTI. $N_p$ and $N_r$ represent the object queries number from proposal and random generation, respectively.

| $N_p$ | $N_r$ | Car | Cyc |
|-------|-------|-----|-----|
| 120 | 0 | 0.835 | 0.703 |
| 180 | 0 | **0.859** | **0.726** |
| 240 | 0 | 0.853 | 0.698 |
| 180 | 50 | 0.844 | 0.687 |

jects. Existing methods, such as Deformable-DETR, SEED, and TransFusion, confirm this notion by demonstrating that an appropriate number of queries enhances model performance across varying scenarios. In practical applications, we design the number of queries based on the object count distribution to improve robustness and reduce sensitivity. For instance, we set 180 queries for the KITTI dataset and 400 queries for the Waymo Sun et al. (2020) dataset, reflecting their differing object counts (see Table 13 and the main paper). The performance of PTN on the occluded scenarios (Table 7 in the main paper) also demonstrates the robustness of our PTN.

In addition, we evaluate the proposal and random queries on the nuScenes Caesar et al. (2020) dataset. We first analyze the distribution of object counts in the nuScenes dataset and calculate the 30%, 60%, and 90% percentiles, which are approximately 40, 60, and 100, respectively. Subsequently, we categorize the estimated object count $CNT$ into three intervals: $[0, 40)$, $[40, 60)$, and $[60, 100]$ according to the object distribution statistics

Table 14: Performance on the nuScnes. $N_q$ and $N_r$ represent the number of proposal queries and random queries, respectively.

| $N_p$ | $N_r$ | NDS | mAP |
|-------|-------|-----|-----|
| - | - | 44.55 | 36.84 |
| 240 | 0 | 44.84 | 36.93 |
| 300 | 0 | 45.85 | 37.11 |
| 400 | 0 | 44.49 | 36.41 |
| 300 | 100 | **47.81** | **38.93** |

of nuScenes. The number of queries is then set to $K_1 = 180$, $K_2 = 240$, and $K_3 = 300$ for these intervals, respectively, following the same strategy as in Waymo.

The results presented in Table 14 demonstrate that the adaptive query number method ($N_p$ = 300) outperforms the baseline, validating its robustness on the nuScenes dataset.

**The design of adaptive K.** We replace our discretized method with a dense (without discretized intervals) prediction strategy. Results in Table 15 show that the estimated object count becomes more accurate when using the dense strategy, but the detection performance decreases (from 64.45 mAP(L2) to 64.11 mAP(L2)). This is because the model tends to overemphasize the adaptive $K$ estimation branch, which interferes with the optimization of the detector.

Table 15: Performance with different object number estimators. MAE represents the mean absolute error of the estimated object count.

| Methods | mAP/mAPH(L2) | MAE |
|---|---|---|
| with discretized intervals (ours) | 64.45/61.57 | 6 |
| without discretized intervals | 64.11/60.97 | 3.9 |

**The number of sampled keys per query.** For each query, we sample $g \times g$ keys. The results are presented in Table 16. With increasing number of the sampled keys, the detection performance of PTN can be consistently improved. However, the corresponding computational costs are also increasing due to more sampled features being performed for query interaction, leading to more latency. Therefore, in our paper, we choose a proper 5×5 as default to trade off the detection performance and latency.

Table 16: Performance on the number of proposal transformer decoder layers on Waymo.

| Grids | mAP/mAPH (L2) | Latency(ms) |
|---|---|---|
| 3×3 | 61.00/60.6 | 55.2 |
| 5×5 | 64.45/61.57 | 56.8 |
| 7×7 | 64.52/61.61 | 59.3 |

**The number of the proposal transformer decoder layer.** As shown in Table 17, we evaluate the performance of PTN on different proposal transformer decoder layers. When only one proposal transformer decoder layer is applied, PTN achieves a relatively poor performance with 61.0/60.6 mAP/mAPH at L2. As the number of layers increases, the performance improves. In this paper, we set the number of proposal transformer decoder layers to 6 to achieve better performance.

Table 17: Performance and cost on Waymo.

| Layers | 3D AP/APH (L2) Veh | Ped | Cyc | mAP/mAPH (L2) | Memory(M) | FLOPs(G) |
|---|---|---|---|---|---|---|
| 1 | 59.7/61.1 | 63.6/59.3 | 59.7/61.3 | 61.0/60.6 | 2021 | 23.5 |
| 3 | 59.4/60.9 | 63.1/59.2 | 62.5/62.4 | 61.3/60.8 | 2025 | 39.5 |
| 6 | 61.9/61.4 | 65.6/58.6 | 65.8/64.5 | 64.4/61.5 | 2027 | 63.4 |

**The efficiency of the CPRM.** We illustrate the accuracy improvements of PTN over Voxel-RCNN at various distance ranges in Figure 7. Firstly, PTN showed significant improvements over Voxel-RCNN on the pedestrian category, where the size of the vehicle is 3 times larger than that of the pedestrian and there are a few points inside the pedestrians. This highlights the importance of capturing fine-grained information (such as high-frequency edges and local boundary variation) for accurately detecting small objects. Furthermore, PTN achieved larger performance gains on distant objects compared with objects closer to the LiDAR sensor across all three categories (especially for the pedestrian, as the point cloud distributions within pedestrian instances are typically sparse and irregular). We believe this is because distant objects with fewer point clouds require more contextual information for accurate detection. Overall, these results demonstrate the effectiveness of our proposed method in detecting small and distant objects.

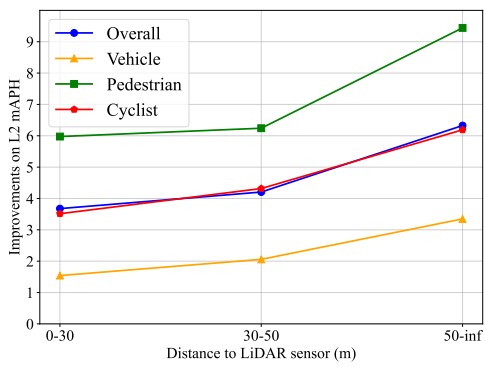

Figure 7: Performance improvement on PTN compared with baseline on Waymo.

**Computational complexity of CPRM.** We evaluate the computational complexity on an NVIDIA GeForce RTX 4090 GPU with a batch size of 1. Results in Table 18 indicate that deformable cross-attention reduces memory costs by approximately fivefold compared to standard cross-attention, while the FLOPs are slightly higher due to the bilinear interpolation operation.

Table 18: Computation complexity of the deformable and standard cross-attention.

| Attention | FLOPs(G) | Memory(MiB) |
|---|---|---|
| deformable cross-attention | 63.4 | 2995 |
| standard cross-attention | 40.4 | 16729 |

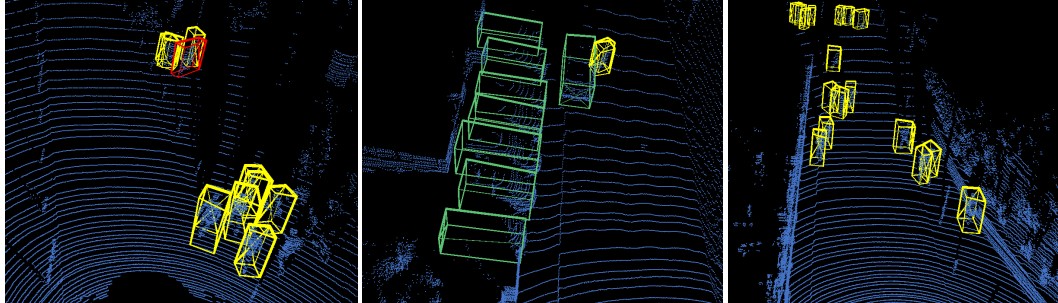

Figure 8: The visualization results using PTN on the KITTI. The black points represent the point clouds, with yellow boxes, green boxes, and red boxes represent pedestrians, vehicles, and cyclists respectively. This visualization demonstrates that our PTN can accurately detect objects in the scene.

**Impact of RPN proposals on the object queries.** The quality of RPN proposals influences the convergence speed and performance of the detector. Specifically, we use the high-quality proposals to initialize potential objects (object queries), which have been proven in prior works Liu et al. (2024); Bai et al. (2022) to be beneficial for accelerating convergence in DETR-like frameworks. To evaluate this effect, we replace the RPN proposals with zeros-initialized queries (where the RPN proposals are extremely poor) following the standard setting of vanilla DETR. The results shown in Table 19 indicate that short schedule training with proposal-initialization achieves better performance than long schedule training with zeros-initialized queries.

Table 19: Performance on Waymo with different queries initialization.

| Methods | mAP/mAPH(L2) |
|---|---|
| proposal-initialization (12 epoch) | 64.45/61.57 |
| zeros-initialization (100 epoch) | 61.23/58.51 |

**Comparison with FSD and FSDv2.** When using the same sampling interval as input, PTN achieves better performance than FSD Fan et al. (2022b) as shown in Table 20. While PTN delivers performance comparable to FSDv2, we attribute the performance of FSDv2 Fan et al. (2024) to its introduction of virtual voxels into the detection head, which improves the mAP L2 by 2.7%. This plug-and-play technique could be integrated into the PTN to achieve further improvements. On the other hand, when trained with fewer data (i.e., at a sample interval of 2), PTN still achieves better performance than FSD.

Table 20: Comparison with other SOTA methods on Waymo.

| Methods | sample interval | mAP/mAPH(L2) |
|---|---|---|
| FSD Fan et al. (2022b) | 1 | 72.9/70.8 |
| FSDv2 Fan et al. (2024) | 1 | 75.6/73.5 |
| PTN | 2 | 73.5/71.2 |
| PTN | 1 | 75.1/72.8 |

**The Influence of Different Coordinates on CPRM.** To better model occlusion relationships, we enhance the positional encoding by incorporating polar coordinates alongside the Cartesian features on CPRM: PE = MLP($x$, $y$, $z$, $l$, $h$, $w$, $\theta$, $\rho$, $\phi$), where $\rho = \sqrt{x^2 + y^2}$ represents the radial distance from the sensor and $\phi = \mathrm{atan2}(y, x)$ is the azimuth angle. The $\rho$ explicitly establishes depth ordering from the sensor's perspective, enabling direct modeling of occlusion: when two objects overlap in angular and height coordinates ($\phi$, $z$), the object with smaller $\rho$ occludes the one with larger $\rho$. Experimental results in Table 21 show that CPRM with polar coordinates achieves a mAP/mAPH (L2) of 64.73/61.82, outperforming the baseline (60.00/57.45) and the Cartesian-only CPRM (64.45/61.57). The improvement is particularly notable for pedestrian and cyclist classes, where occlusion handling is critical. This demonstrates that polar coordinates more naturally represent occlusion relationships and reduce model learning complexity.

Table 21: Performance with different coordinates on Waymo.

| Methods | 3D AP/APH (L2) | | |
|---|---|---|---|
| | Veh | Ped | Cyc |
| Baseline | 59.09/58.57 | 58.20/52.47 | 62.73/61.33 |
| CPRM(Cartesian) | **61.91/61.49** | 65.62/58.69 | 65.82/64.54 |
| CPRM (Polar) | 61.87/61.45 | **65.77/58.84** | **66.56/65.17** |

**The visualization results.** The visualization results using our PTN on KITTI Geiger et al. (2012) are shown in Figure 8. From the results, we can see that our PTN can detect most objects, especially for small objects (such as pedestrians). This is attributed to the HAFA module, which efficiently retrieves the lost detail information in the downsampling process. This information is important for the detector to accurately locate small objects.

