# OpenReview forum: "PTNET: A PROPOSAL-CENTRIC TRANSFORMER NET- WORK FOR 3D OBJECT DETECTION"
_ICLR.cc/2026/Conference — ICLR 2026 Poster_

### Official Review · Reviewer_Hw6U · 2025-10-30

**Soundness:** 2
**Presentation:** 3
**Contribution:** 2
**Rating:** 4
**Confidence:** 4

**Summary:**

The paper tackles the proposal-quality bottleneck in two-stage LiDAR 3D detectors by introducing PTN, a proposal-centric transformer composed of two key parts: Hierarchical Attentive Feature Alignment (HAFA) and a Collaborative Proposal Refinement Module (CPRM). HAFA is a dual-stream design that fuses coarse, multi-scale voxel features with fine, foreground point features and then aligns them in a unified space to strengthen geometric detail in proposals. CPRM creates hybrid object queries—top-K proposal queries plus learnable random queries—and performs 3D parameter-guided deformable attention across proposals to share contextual cues, especially helpful under occlusion and sparsity. Experiments on Waymo and KITTI report consistent gains; e.g., Waymo test mAP/mAPH (L2) of 72.7/70.6 with strong category results, and competitive KITTI performance, alongside ablations justifying each component.

**Strengths:**

1. Clear problem framing: identifies two concrete issues—loss of geometric detail in proposals and lack of cross-proposal context—and directly designs modules to address each.
2. Well-motivated dual-stream features: combining grid-sampled voxel tokens (coarse) with raw foreground point cues (fine) is intuitive and technically grounded, with an explicit alignment step.
3. Ablations support claims: both HAFA and CPRM contribute; component-wise tables and studies on query counts/NMS thresholds/decoder depth are informative.

**Weaknesses:**

1. Added complexity & runtime: HAFA + CPRM introduce nontrivial overhead and multiple hyperparameters (e.g., grid sizes, query counts, NMS/thresholds); the paper discusses trade-offs qualitatively but detailed compute/memory costs and scaling behavior are limited.
2. RPN dependence: although CPRM adds random queries, the pipeline still leans on RPN quality; failure modes when RPN proposals are poor (domain shift, long-range sparse objects) are not deeply analyzed.

**Questions:**

1. Deformable attention specifics: How are reference points and offsets parameterized from 3D boxes? What’s the exact number of sampled keys per query and the computational/memory cost per decoder layer?
2. Query budgeting: Beyond global K, did you try class-aware (per-class) query budgets or adaptive K per scene without discretized intervals? How sensitive is performance to K and the random-query count M across datasets?

---

> ### Author Response · Authors · 2025-11-28
> **Response to reviewer Hw6U.(1)**
>
> Dear Reviewer Hw6U,
>
> Thank you for your detailed comments and suggestions. We tried our best to address all the concerns and questions, and update the main paper and appendix(marked blue) in the new version. Please let us know if you have any further concerns or questions to discuss. ﻿
>
> Best, ﻿
>
> Paper 22410 Authors
>
> **Q1**: ***Complexity and memory cost of HAFA and CPRM.***
>
> **A1**: We evaluate the computational complexity on an NVIDIA GeForce RTX 4090 GPU with a batch size of 1. The results are shown in Table1 below. To analyze the __scaling behavior__ of PTN, we conduct experiments on two key hyperparameters: the query counts and the grid size. As shown in table2 below , the performance remains stable across different query counts. Additionally, we report the latency for various grid sizes. The results shown in Table3 below demonstrate that the overall latency of PTN is largely insensitive to changes in the grid size.
>
> **Table1:Inference cost of each component**
>
> | HAFA | CPRM | mAP/mAPH (L2) | Memory (MiB) | FLOPs (G) |
> |------|------|---------------|--------------|-----------|
> |      |      | 60.00/57.45   | 4533         | 101.8     |
> | ✓    |      | 63.52/60.19   | 2431         | 21.7      |
> |      | ✓    | 64.45/61.57   | 2027         | 63.4      |
> | ✓    | ✓    | **66.40/63.58** | 2995       | **69.6**  |
>
> **Table2:Performance on the nuScenes**
> *$N_p$ and $N_r$ represent the number of proposal queries and random queries, respectively. The first row represents the baseline*
>
> | $N_p$| $N_r$ | NDS   | mAP  |
> |-----|-----|-------|------|
> | -   | -   | 44.55 | 36.84 |
> | 240 | 0   | 44.84 | 36.93 |
> | 300 | 0   | 45.85 | 37.11 |
> | 400 | 0   | 44.49 | 36.41 |
> | 300 | 100 | **47.81** | **38.93** |
>
> **Table3:Performance on the number of proposal transformer decoder layers on Waymo.**
>
> | Grids | mAP/mAPH (L2) | Latency (ms) |
> |-------|---------------|--------------|
> | 3×3   | 61.00/60.6    | 55.2         |
> | 5×5   | 64.45/61.57   | 56.8         |
> | 7×7   | 64.52/61.61   | 59.3         |
>
>
> **Q2**: ***Impact of RPN proposals on the object queries.***
>
> **A2**: The quality of RPN proposals influences the convergence speed and performance of the detector. Specifically, we use the high-quality proposals to initialize potential objects (object queries), which have been proven in prior works to be beneficial for accelerating convergence in DETR-like frameworks. To evaluate this effect, we replace the RPN proposals with zeros-initialized queries (where the RPN proposals are extremely poor) following the standard setting of vanilla DETR. The results shown in the table4 below indicate that short schedule training with proposal-initialization achieves better performance than long schedule training with zeros-initialized queries.
>
> **Table4:Performance on Waymo with different queries initialization**
>
> | Methods | mAP/mAPH (L2) |
> |---------|---------------|
> | proposal-initialization (12 epoch) | 64.45/61.57 |
> | zeros-initialization (100 epoch) | 61.23/58.51 |
>
> **Q3**: ***The Design of Deformable Attention (DA).***
>
> **A3**: The detailed structure of the Deformable Box Attention (DBA) module is shown in Appendix A, Figure 5. The core idea of our deformable attention mechanism is to replace the reference point with a reference box, enabling more comprehensive geometric modeling. This box-based representation provides richer spatial information than point-based approaches, which significantly enhances both regression and classification performance.
>
> Specifically, given a proposal $b_i = (x, y, z, l, h, w, θ)$, where (x, y, z) denotes the center coordinates, (l, h, w) represents the dimensional size, and θ indicates the orientation, we first normalize its coordinates to the range [0, 1] relative to the point cloud range $(x_0, y_0, z_0, x_1, y_1, z_1)$ as follows:
> $br=[(x-x_0)/(x_1-x_0), (y-y_0)/(y_1-y_0), (z-z_0)/(z_1-z_0), l/(x_1-x_0), h/(y_1-y_0), w/(z_1-z_0), (θ+π)/2π]$
>
> Subsequently, we compute the offset and regression output through the following steps:
>
> $offset = MLP(DA(F_k, F + MLP(br)))
> out = σ(offset + inverse\\_sigmoid(br))$
>
> where σ denotes the sigmoid activation function and inverse_sigmoid(·) represents the inverse sigmoid transformation that maps values from [0,1] back to the original domain.
>
> This design enables the model to learn adaptive sampling locations around the reference box, allowing it to capture features from relevant neighboring regions. The box-aware formulation provides two key advantages: (1) it preserves the complete geometric structure of proposals, including size and orientation information; (2) it facilitates more accurate feature aggregation by considering the spatial extent of objects rather than just their centers.
>
> The learned offsets allow the attention mechanism to dynamically adjust sampling locations based on both the proposal features and geometric properties, leading to more effective cross-proposal interactions and ultimately stronger proposal refinement.

---

> ### Author Response · Authors · 2025-11-28
> **Response to reviewer Hw6U.(2)**
>
> **Q4**: ***The number of sampled keys per query.***
>
> **A4**: For each query, we sample g × g keys. The results are presented in the table5 below. With increasing number of the sampled keys, the detection performance of PTN can be consistently improved. However, the corresponding computational costs are also increasing due to more sampled features being performed for query interaction, leading to more latency. Therefore, in our paper, we choose a proper 5×5 as default to trade off the detection performance and latency.
>
> **Table5:Performance on the number of proposal transformer decoder layers on Waymo.**
>
> | Grids | mAP/mAPH (L2) | Latency (ms) |
> |-------|---------------|--------------|
> | 3×3   | 61.00/60.6    | 55.2         |
> | 5×5   | 64.45/61.57   | 56.8         |
> | 7×7   | 64.52/61.61   | 59.3         |
>
> **Q5**: ***Computational and memory costs of each decoder.***
>
> **A5**: The computational cost shown in the table6 below indicates that FLOPs increase as the number of layers grows, while the memory remains stable because the transformer decoder is relatively small compared to the detector.
>
> **Table6:Performance on the number of proposal transformer decoder layers on Waymo.**
>
> | Layers | mAP/mAPH (L2) | Memory (MiB) | FLOPs (G) |
> |--------|---------------|--------------|-----------|
> | 1      | 61.0/60.6     | 2021         | 23.5      |
> | 3      | 61.3/60.8     | 2025         | 39.5      |
> | 6      | 64.4/61.5     | 2027         | 63.4      |
>
> **Q6**: ***The design of adaptive K.***
>
> **A6**: We replace our discretized method with a dense (without discretized intervals) prediction strategy. Results in the table below show that the estimated object count becomes more accurate when using the dense strategy, but the detection performance decreases (from 64.45 mAP(L2) to 64.11 mAP(L2)). This occurs because the model tends to overemphasize the adaptive K estimation branch, which interferes with the detector's optimization. Additionally, the dense strategy is sensitive to this value, resulting in a high standard deviation in the estimated object count. Consequently, an underestimated count can easily cause the model to overlook objects, ultimately leading to a drop in recall.
>
> **Table7:Performance with different object number estimators. MAE represents the mean absolute error of the estimated object count.**
>
> | Methods | mAP/mAPH (L2) | MAE |
> |---------|---------------|-----|
> | with discretized intervals (ours) | 64.45/61.57 | 6 |
> | without discretized intervals | 64.11/60.97 | 3.9 |
>
> **Q7**: ***Robustness with respect to proposal number K and random query number M.***
>
> **A7**: We use K = 300 and M = 100 as the default setting for the Waymo, nuScenes, and KITTI datasets. Results in the table below demonstrate that, under this default configuration, PTN achieves better performance than the baseline.
>
> **Table8:Performance across different datasets with the numbers of proposal and random queries set to K = 300 and M = 100, respectively.**
>
> | Dataset | Waymo | nuScenes | KITTI |
> |---------|-------|----------|-------|
> | Voxel R-CNN | 59.1/55.1 | 44.5 | 74.1 |
> | PTN | 64.4/61.5 | 47.8 | 76.5 |

---

### Official Review · Reviewer_1kqr · 2025-10-31

**Soundness:** 3
**Presentation:** 3
**Contribution:** 3
**Rating:** 6
**Confidence:** 4

**Summary:**

This paper proposes a Proposal-centric Transformer Network (PTN) for 3D object detection from LiDAR point clouds. PTN addresses two main challenges in two-stage 3D detection: degradation of proposal geometric details due to point sparsity and ineffective use of contextual cues during refinement. PTN introduces two core modules: Hierarchical Attentive Feature Alignment (HAFA), which extracts and aligns coarse voxel and fine point features within each proposal, and Collaborative Proposal Refinement Module (CPRM), which integrates contextual interaction among spatially and semantically related proposals via hybrid queries and deformable attention. Extensive experiments on the Waymo and KITTI benchmarks demonstrate performance improvements over representative prior methods.

**Strengths:**

1. The Collaborative Proposal Refinement Module is the first proposal-centric transformer to refine the bounding boxes generated from RPN. The idea is interesting, and the performance gain is significant. With the random queries, it can recall weak proposals caused by distance or occlusion.
2. The final performance on the Waymo Open dataset (single-frame setting) is impressive, especially on small objects.
3. Ablation studies isolate the effect of each module, providing transparency on their contributions to accuracy and speed. Table 9 offers inference speed comparisons per module, supporting claims of balanced efficiency and performance.
4. The paper contextualizes itself against a range of competing methods, including both DETR-style and non-Transformer 3D detectors, and references and discusses most major prior works.

**Weaknesses:**

1. One of the effects of CPRM is that it is an end-to-end module. However, the authors still apply NMS before the module, which weaken the end-to-end feature. Moreover, I would like to see a comparison between CPRM and classic RCNN module (e.g. Voxel-RCNN) with NMS.
2. Section 3.3.3 introduces a “3D parameter-guided deformable attention,” but omits critical specifics: how offsets are computed per proposal, how attention ranges are constrained spatially, and what prior (if any) is imposed by the box parameters. A clear symbolic or algorithmic formulation is needed for reproducibility and transparency.
3. The multi-frame performance is not superior to pervious work CenterFormer. Does the proposal collect features from the "tail" caused by object movements?
4. Current state-of-the-art works, such as FSD and FSDv2, are not cited and compared.

**Questions:**

Is the position embedding based on xyz only? How can such position embedding represents "occlusion" between proposals?

---

> ### Author Response · Authors · 2025-11-28
> **Response to reviewer 1kqr.(1)**
>
> Dear Reviewer 1kqr,
>
> Thank you for your detailed comments and suggestions. We tried our best to address all the concerns and questions, and update the main paper and appendix(marked blue) in the new version. Please let us know if you have any further concerns or questions to discuss. ﻿
>
> Best, ﻿
>
> Paper 22410 Authors
>
>
>
> **Q1**: ***End-to-end pipeline.***
>
> **A1**: An ideal, purely end-to-end detection system should avoid any heuristic post-processing steps, such as NMS. Current works, such as the DETR series, are representative of this direction. In contrast, classical two-stage approaches such as Voxel R-CNN commonly employ NMS, yet these methods(such as PV-RCNN, PV-RCNN++, Voxel-RCNN) are still described as "end-to-end training" in their original papers.
>
> For classic two-stage detectors, the use of NMS in the second stage mainly serves to filter out a large number of low-quality proposals. This design has two practical advantages:
>
> - **Computational efficiency**: retaining an excessive number of candidates dramatically increases the cost of subsequent modules
> - **Model performance**: most removed proposals correspond to clear background regions and contribute little to training the RCNN head
>
> In principle, removing NMS entirely would also make our CPRM pipeline strictly end-to-end. However, doing so introduces a non-negligible drop in efficiency and performance. Importantly, incorporating NMS does not break the end-to-end property. The surviving high-quality boxes preserve meaningful gradients, enabling effective back-propagation to the first-stage network.
>
>
>
> **Q2**: ***Comparison with Voxel R-CNN.***
>
> **A2**: The comparison between CPRM and Voxel R-CNN is presented in the table1 below. The results demonstrate that CPRM achieves superior performance on sparse object categories. As described in the main paper (Section 4.2, lines 365-367), Voxel R-CNN is used as the baseline. The detailed comparison results can be found in Table 4 of the main paper (lines 421-427).
>
> **Table1:Ablation study of CPRM on Waymo.**
>
> | Methods | Vehicle 3D AP/APH (L2) | Pedestrian 3D AP/APH (L2) | Cyclist 3D AP/APH (L2) | mAP/mAPH (L2) |
> |---------|------------------------|---------------------------|------------------------|---------------|
> | VR      | 59.09/58.57            | 58.20/52.47               | 62.73/61.33            | 60.00/57.45   |
> | CPRM    | 61.91/61.49            | 65.62/58.69               | 65.82/64.54            | 64.45/61.57   |
>
>
> **Q3**: ***Details of DBA.***
>
> **A3**:The detailed structure of the Deformable Box Attention (DBA) module is shown in Appendix A, Figure 5. The core idea of DBA is to divide each proposal into $G^2$ grid points and learn an offset for every grid, enabling it to aggregate features from neighboring proposals. The grid features are then fused through learnable attention weights to obtain the final interaction-enhanced proposal representation. This grid-based decomposition provides two key benefits: (1) using multiple grid locations offers richer semantics and spatial information than relying solely on the proposal center; (2) the grid layout implicitly encodes proposal size, which strengthens both regression and classification.
>
> Specifically, given a proposal $b_i=(x, y, z, l, h, w, \theta)$, where $(x, y, z)$ denotes the center coordinates, $(l, h, w)$ represents the dimensional size, and $\theta$ indicates the orientation, we first normalize its coordinates to the range [0, 1] relative to the point cloud range $(x_0, y_0, z_0, x_1, y_1, z_1)$ as follows $b^* = [(x-x_0)/(x_1-x_0), (y-y_0)/(y_1-y_0), (z-z_0)/(z_1-z_0), l/(x_1-x_0), h/(y_1-y_0), w/(z_1-z_0), (θ+π)/2π]$
>
> Subsequently, we convert the proposal into a set of $G^2$ grid points $P_g$, as shown in the figure. These grid points are projected onto the proposal voxel feature map $F$ to extract the corresponding grid features $F_g=\\{f_g^k\\}_{k=1}^{G^2}$.
>
> For all grids, we first compute the normalized offset $O_g= \sigma(MLP(F_g))$ from $F_g$, where $\sigma$ denotes the sigmoid function to normalize the offset into [0,1]. The final sampling locations are the normalized offset locations $O_g$ plus the initial grid locations $P_g$.
>
> Based on the sampling locations, we apply bilinear interpolation on the voxel feature map to obtain the sampled grid features $F_g=\\{f_g^k\\}_{k=1}^{G^2}$. The learned offsets allow grids to sample features from nearby proposals, naturally enabling cross-proposal feature interaction.
>
> In parallel, we compute attention weights for all grids: $A = softmax(MLP(F_g))=\\{a_k\\}_{k=1}^{G^2}$.
>
> Finally, the enhanced proposal representation is obtained by a weighted sum over all sampled grid features: $F_p = \sum_{k=1}^{G^2} a_k \cdot f_s^k$.
>
> This design enables rich feature exchange between proposals and implicitly encodes geometric structure, leading to stronger proposal refinement.

---

> ### Author Response · Authors · 2025-11-28
> **Response to reviewer 1kqr.(2)**
>
> **Q4**: ***Comparison with CenterFormer.***
>
> **A4**:As shown in the main paper, PTN achieves comparable performance to CenterFormer while using only three input frames compared to CenterFormer's four. Furthermore, as shown in the table2 below, PTN outperforms CenterFormer when the same number of multi-frame inputs are used.
>
> **Table2:Effectiveness of PTN on Waymo validation set using multi-frame inputs.**
>
> | Methods | Frames | mAP/mAPH (L2) |
> |---------|--------|---------------|
> | CenterFormer | 4 | 74.7/73.2 |
> | PTN | 3 | 74.5/73.2 |
> | PTN | 4 | **75.6/74.1** |
>
> **Q5**: ***Comparison with FSD and FSDv2.***
>
> **A5**: When using the same sampling interval as input, PTN achieves better performance than FSD as shown in the table3 below. While PTN delivers performance comparable to FSDv2, we attribute the performance of FSDv2 to its introduction of virtual voxels into the detection head, which improves the mAP L2 by 2.7%. This plug-and-play technique could be integrated into the PTN to achieve further improvements. On the other hand, when trained with fewer data (i.e., at a sample interval of 2), PTN still achieves better performance than FSD. We will discuss them in the Related Work.
>
> **Table3:Comparison result on Waymo.**
>
> | Methods | Sample Interval | mAP/mAPH (L2) |
> |---------|----------------|---------------|
> | FSD | 1 | 72.9/70.8 |
> | FSDv2 | 1 | 75.6/73.5 |
> | PTN | 2 | 73.5/71.2 |
> | PTN | 1 | 75.1/72.8 |
>
> **Q6**: ***Design of positional embedding.***
>
> **A6**:Our positional encoding goes beyond using only center coordinates (x, y, z); it incorporates the full geometric parameters (x, y, z, l, h, w, θ), where (l, h, w) denote the proposal size and θ denotes its orientation. This richer geometric representation provides the model with explicit cues about the 3D shape and spatial extent of each proposal.
>
> Regarding "occlusion" relationships between proposals, these geometric parameters allow the model to implicitly infer such interactions. Specifically, given the 3D shape of each proposal, the model can reason about relative spatial arrangement and thus estimate whether two proposals overlap. For example, if their 3D extents are far apart or the proposals are very small, the model can infer that occlusion is unlikely. In addition, depth differences along the x-axis naturally encode front-back ordering, enabling the model to identify when a more distant proposal is likely to be occluded by a closer one.
>
> Beyond geometry alone, combining these geometric embeddings with proposal features that capture content-level cues (etc., point cloud density, missing regions, and structural incompleteness) further strengthens the model's ability to learn occlusion patterns during training. In other words, geometric parameters (x, y, z, l, h, w, θ) provide a coarse estimate of potential overlap, while proposal features reflecting incomplete or sparse point cloud observations provide strong evidence for actual occlusion.
>
> Together, these complementary cues allow the model to reliably learn occlusion relationships between proposals without requiring explicit occlusion annotations.

---

> > ### Comment · Reviewer_1kqr · 2025-11-28
> >
> > I appreciate the authors' detailed response. Most of my concerns have been solved and I have seen the newly added Details of PTN in Appendix A.
> >
> > Now I still have two concerns about this paper.
> >
> > 1. The explanation on the design of positional embedding cannot persuade me. I am wondering if polar or cylinder coordinates could be better choices for position embedding, under which the occlusion can be directly computed.
> >
> > 2. In the updated paper, I still cannot see the citation and comparison with FSD and FSDv2.

---

> ### Author Response · Authors · 2025-11-28
> **Response to reviewer 1kqr.(3)**
>
> **Q1**:  ***Design of positional embedding.***
>
> **A1**: We thank the reviewer for their insightful comment. Indeed, compared to the Cartesian coordinate system, using a cylinder coordinate system can more naturally represent occlusions, explicitly model occlusion relationships, and reduce the learning complexity of the model. In contrast, the Cartesian coordinate system implicitly represents these relationships, which to some extent increases the model's learning complexity. The cylinder coordinate system is similar to the range view and more fundamentally captures spatial relationships.
>
> Specifically, we plan to retain all geometric parameters of the bounding box, including the center coordinates $(x, y, z)$, dimensions $(l, h, w)$, and orientation angle $\theta$, while simultaneously converting the center point coordinates from Cartesian coordinates to cylinder coordinates $(\rho, \phi, z)$. The conversion formulas are:
>
> $$
> \rho = \sqrt{x^2 + y^2}, \quad
> \phi = \arctan\left(\frac{y}{x}\right), \quad
> z = z.
> $$
>
> This approach enables direct modeling of occlusion because the radial distance $\rho$ directly establishes a depth order from the sensor's perspective: when two objects overlap in the angular and height coordinates $(\phi, z)$, the object with the smaller $\rho$ value will inevitably occlude the one with the larger $\rho$ value. This representation allows the model to explicitly utilize depth information for occlusion reasoning, rather than implicitly learning it from Cartesian coordinates.
>
> We plan to incorporate the cylinder coordinates $(\rho, \phi, z)$ as new positional encoding features in the revised version, and input them together with all existing geometric parameters (including the Cartesian center coordinates, dimensions $(l, h, w)$, and orientation angle $\theta$) into the CPRM, to enhance the model's ability to understand complex occlusion scenarios. We will commence the evaluation of the results for CPRM promptly.
>
> We thank the reviewer again for this constructive comment.
>
> **Q2**:  ***Comparison with FSD and FSDv2.***
>
> **A2**:  We apologize for this oversight. In the revised manuscript, we have addressed this concern by adding comprehensive comparisons in Appendix C (Table 20) that directly compare PTN with both FSD and FSDv2, and including proper citations in the related work section (lines 112-113 and line 119-121).
>
> [1]Fan L, Wang F, Wang N, et al. Fully sparse 3d object detection[J]. Advances in Neural Information Processing Systems, 2022, 35: 351-363.
>
> [2] Fan L, Wang F, Wang N, et al. Fsd v2: Improving fully sparse 3d object detection with virtual voxels[J]. IEEE Transactions on Pattern Analysis and Machine Intelligence, 2024.

---

> ### Author Response · Authors · 2025-11-30
> **Response to reviewer 1kqr.(4)**
>
> **Q1**:  ***Design of positional embedding.***
>
> **A1**:We sincerely thank the reviewer for the insightful comment and constructive feedback. Following the suggestion, we thoroughly investigate using polar coordinates (essentially cylinder coordinates in LiDAR perception) for positional embedding in our CPRM module.
>
> In our original implementation, CPRM uses a Multi-Layer Perceptron (MLP) to encode the geometric parameters of bounding boxes: $PE = MLP(x, y, z, l, h, w, \theta)$. Based on the valuable suggestion, we now enhance this positional encoding by incorporating polar coordinates alongside the original Cartesian features: $PE = MLP(x, y, z, l, h, w, \theta, \rho, \phi)$, where $\rho = \sqrt{x^2 + y^2}$ represents the radial distance from the sensor, $\phi = \text{atan2}(y, x)$ is the azimuth angle computed using the two-argument arctangent function.
>
> We specifically use $\text{atan2}(y, x)$ instead of $\arctan(y/x)$ to properly handle all quadrants and avoid division-by-zero issues, which is crucial for robust coordinate transformation.
>
> The experimental results on the Waymo dataset demonstrate that this enhanced positional encoding with polar coordinates brings consistent improvements. As shown in Table 1, CPRM with polar coordinates achieves a mAP/mAPH (L2) of 64.73/61.82, outperforming both the baseline (60.00/57.45) and the Cartesian-only CPRM (64.45/61.57). The improvement is particularly notable for pedestrian and cyclist, where occlusion handling is more critical. This validates the intuition that polar coordinates can more naturally represent occlusion relationships. Specially, the radial distance $\rho$ explicitly establishes depth ordering from the sensor's perspective, enabling the model to better reason about which objects occlude others when they overlap in angular space.
>
> We update our manuscript to include these findings and thank the reviewer again for this valuable suggestion, which significantly strengthens our work.
>
> **Table 1: Performance with different coordinates on Waymo.**
>
> | Methods             | Vehicular AP/APH (L2) | Pedestrian AP/APH (L2) | Cyclist AP/APH (L2) | Overall mAP/mAPH (L2) |
> |---------------------|------------------------|-------------------------|----------------------|------------------------|
> | Baseline            | 59.09/58.57           | 58.20/52.47            | 62.73/61.33         | 60.00/57.45           |
> | CPRM (Cartesian)    | **61.91/61.49**           | 65.62/58.69            | 65.82/64.54         | 64.45/61.57           |
> | CPRM (Polar)        | 61.87/61.45           | **65.77/58.84**        | **66.56/65.17**     | **64.73/61.82**       |
>
>
> **Q2**:  ***Comparison with FSD and FSDv2.***
>
> **A2**:  We apologize for this oversight. In the revised manuscript, we have addressed this concern by adding comprehensive comparisons in __Appendix C (Table 20)__ that directly compare PTN with both FSD and FSDv2, and including proper citations in the related work section (__lines 112-113 and line 119-121__).
>
> [1]Fan L, Wang F, Wang N, et al. Fully sparse 3d object detection[J]. Advances in Neural Information Processing Systems, 2022, 35: 351-363.
>
> [2] Fan L, Wang F, Wang N, et al. Fsd v2: Improving fully sparse 3d object detection with virtual voxels[J]. IEEE Transactions on Pattern Analysis and Machine Intelligence, 2024.

---

### Official Review · Reviewer_uUA1 · 2025-10-31

**Soundness:** 3
**Presentation:** 2
**Contribution:** 2
**Rating:** 4
**Confidence:** 3

**Summary:**

This paper introduces PTNET, a novel Proposal-centric Transformer Network designed to enhance two-stage 3D object detection from LiDAR point clouds. PTNET targets two key limitations of current two-stage methods: 1) the degradation of geometric details in proposal features due to point cloud sparsity and pooling operations; and 2) the failure to leverage contextual clues from neighboring proposals during the refinement stage, which traditionally treats each proposal independently. Experimental results demonstrate that PTNET achieves state-of-the-art (SOTA) performance on the Waymo and KITTI large-scale benchmarks.

**Strengths:**

- The paper is well-motivated, introducing DETR into the second stage of 3D detectors to aggregate information from the full ROIs
- The authors conduct extensive experiments on two major autonomous driving benchmarks (Waymo and KITTI), demonstrating state-of-the-art or highly competitive performance across multiple categories, especially for pedestrians and cyclists. The multi-frame input results further validate the model's robustness.

**Weaknesses:**

- While the combination of HAFA and CPRM is well executed, the individual ideas (multi-granularity alignment and proposal interaction through attention) resemble previous works such as PV-RCNN++[1] and ConQueR[2]. The contribution may thus be seen as an engineering refinement rather than a conceptual breakthrough.
- The fine-grained branch (FPFR) in HAFA is meant to recover geometric detail from "raw foreground point clouds." As described in Section 3.2.2, this module "first selects foreground points $\text{P}'$ whose locations are inside the proposal $b$." This introduces a limitation: if an RPN proposal $b$ is poor (e.g., a tiny bounding box due to heavy occlusion), FPFR can only access points within that incomplete proposal. It can therefore only "sharpen" the features within the existing, poor boundary, but seems inherently unable to "complete" the true geometric shape by accessing points that belong to the object but lie outside the RPN's initial prediction.

[1] Shi, Shaoshuai, et al. "PV-RCNN++: Point-voxel feature set abstraction with local vector representation for 3D object detection." International Journal of Computer Vision 131.2 (2023): 531-551.
[2] Zhu, Benjin, et al. "Conquer: Query contrast voxel-detr for 3d object detection." Proceedings of the IEEE/CVF Conference on Computer Vision and Pattern Recognition. 2023.

**Questions:**

- How robust is the adaptive query number estimation (Eq. 5–6) to datasets with different object count distributions, such as nuScenes or Argoverse?
- The CPRM employs deformable cross-attention among proposals. What is its computational complexity compared with standard cross-attention, and how does it scale when the number of proposals increases?

---

> ### Author Response · Authors · 2025-11-28
> **Response to reviewer uUA1.(1)**
>
> Dear Reviewer uUA1,
>
>
> Thank you for your detailed comments and suggestions. We tried our best to address all the concerns and questions, and update the main paper and appendix(marked blue) in the new version. Please let us know if you have any further concerns or questions to discuss.
> ﻿
>
> Best,
> ﻿
>
> Paper  22410 Authors
>
> __Q1__:The difference compared with PV-RCNN++ and ConQueR.
>
> __A1__:While existing approaches, such as PV-RCNN++, introduce fine-grained point features, and ConQueR introduces proposal interaction, they exhibit limitations in efficiently enhancing proposal representation. In contrast, our method introduces several innovative components that significantly enhance the utilization of fine-grained point features (HAFA module) and proposal interaction (CPRM module) to enhance proposal representation:
>
> **1. Innovation in Fine-Grained Point Feature Utilization.**
>
> - **Challenge:** Existing approaches, particularly PV-RCNN++, adopt the Furthest Point Sampling (FPS) algorithm to sample a small number of keypoints P* from the raw point cloud P and construct fine-grained point features based on P*. These point features are then concatenated with voxel features. However, the sampling process inevitably loses some foreground points, which compromises the recall, especially for small objects (e.g., cyclists). As reported in IA-SSD, the instance recall rate for the cyclist drops to 97.2% after the sampling process. Moreover, the simple concatenation operation is insufficient to align features of different granularities.
> - **Our Improvement:**
>   - We recover fine-grained features directly from the raw point cloud P without the sampling process to avoid the loss of foreground points.
>   - We employ a semantic attention network in HAFA to align fine-grained point features with coarse-grained voxel features.
>
> **2. Architecture-Level Innovations.**
>
> - **Challenge:** ConQueR uses a fixed k to select proposals as object queries. However, this design negatively impacts the detector's recall when the number of objects exceeds k.
> - **Our Improvement:** In CPRM, we introduce an object count estimation network to adaptively estimate k for each scene, which provides a prior for the detector to ensure high recall. Additionally, we incorporate random queries to retrieve some overlooked objects by proposal queries, further improving the recall.
> - **Challenge:** ConQueR uses the center features and proposal features as the key and query in the proposal interaction, which lacks the utilization of geometric priors.
> - **Our Improvement:**
>   - We use the full 7D proposal boxes as keys (and values), not just 2D centers. This provides a *geometrically aligned* and information-rich context for every cross-attention step. The network can perform feature matching and refinement based on comprehensive cues (e.g., "adjust orientation to match that nearby box of similar size"), leading to more precise box adjustments.
>   - While ConQueR's query-center interaction primarily enriches feature representation, our proposal-to-proposal interaction performs *direct geometric refinement*. Proposals can mutually adjust their coordinates, dimensions, and angles by attending to their neighbors, effectively performing a consensus-based optimization. This leads to higher IoU for the final boxes and a more stable detection output, as outliers are corrected by spatially coherent predictions.
>
> **3. Efficiency Improvements.**
>
> - **Challenge:** ConQuER employs a denoising branch to accelerate convergence and improve performance. However, this introduces additional training costs that scale linearly with the number of denoising groups.
> - **Our Solution:** We introduce random queries as an additional complement to the proposal queries, which efficiently reduces training costs.
>
> These innovations work synergistically to significantly advance different detectors (PV-RCNN++: +6.2% mAPH (L2), ConQueR: +3.5% mAPH (L2)), demonstrating substantial improvements on the Waymo dataset.
>
> [1] Shi, Shaoshuai, et al. "PV-RCNN++: Point-voxel feature set abstraction with local vector representation for 3D object detection." International Journal of Computer Vision 131.2 (2023): 531-551.
>
> [2] Zhu, Benjin, et al. "Conquer: Query contrast voxel-detr for 3d object detection." Proceedings of the IEEE/CVF Conference on Computer Vision and Pattern Recognition. 2023.
>
> [3] Zhang, Y., et al. "Not all points are equal: Learning highly efficient point- based detectors for 3d lidar point clouds." Proceedings of the IEEE/CVF Conference on Computer Vision and Pattern Recognition. 2022.

---

> ### Author Response · Authors · 2025-11-28
> **Response to reviewer uUA1.(2)**
>
> **Q2**: ***Impact of the poor RPN proposals.***
> **A2**:During training, only high-quality RPN proposals are used to "complete" the true geometric shape. Specifically, for a proposal b, we first extract its interior raw point clouds P' and then obtain the point cloud features F via PointNet. If the IoU between b and a ground-truth box exceeds 0.5, F is used for regression (i.e., "complete" the true geometric shape). Otherwise, for low-quality (poor) RPN proposals, F is used solely for background classification.
>
> We address the issue of poor RPN proposals that produce tiny bounding boxes due to heavy occlusion from two aspects. First, we introduce a feature alignment module within HAFA to refine incomplete local point features. This module employs a gated architecture to balance voxel and point features.  When point features are unreliable, it prioritizes voxel features to enhance the overall representation. Second, we apply the CPRM to model interactions among proposals with incomplete geometric shapes using deformable box attention.  This enables the model to effectively represent objects that are split into multiple parts.
>
>
> **Q3**: ***Robustness of query number on nuScenes.***
> **A3**:Based on Eq. (5) and (6), we analyze the distribution of object counts in the nuScenes dataset and calculate the 30%, 60%, and 90% percentiles (see Appendix B), which are approximately 40, 60, and 100, respectively. We then categorize the estimated object count CNT into three intervals: [0,40), [40,60), and [60,100], based on these distribution statistics. Following the same strategy as used for Waymo, we set the number of queries to K1 = 180, K2 = 240, and K3 = 300 for these intervals, respectively. The results in the table1 below demonstrate that our adaptive query method (Np = 300) outperforms the baseline, validating the robustness of query number on the nuScenes dataset.
>
> **Table1:Performance on the nuScenes**
> *$N_p$ and $N_r$ represent the number of proposal queries and random queries, respectively. The first row represents the baseline*
>
> | $N_p$  | $N_r$  | NDS   | mAP  |
> |-----|-----|-------|------|
> | -   | -   | 44.55 | 36.84 |
> | 240 | 0   | 44.84 | 36.93 |
> | 300 | 0   | 45.85 | 37.11 |
> | 400 | 0   | 44.49 | 36.41 |
> | 300 | 100 | **47.81** | **38.93** |
>
>
>
> **Q4**: ***Computational complexity of CPRM.***
> **A4**:We evaluate the computational complexity of CPRM on an NVIDIA GeForce RTX 4090 GPU with a batch size of 1. The results in the table2 below indicate that deformable cross-attention reduces memory costs by approximately fivefold compared to standard cross-attention, while the FLOPs are slightly higher due to the bilinear interpolation operation.
>
> **Table2: Computation complexity of the deformable and standard cross-attention**
>
> | Attention | FLOPs(G) | Memory(MiB) |
> |-----------|----------|-------------|
> | deformable cross-attention | 63.4 | 2995 |
> | standard cross-attention | 40.4 | 16729 |
>
>
>
> Furthermore, we analyze the computational cost of CPRM as the number of proposals increases as shown in the table3 below. The results demonstrate that deformable cross-attention is more efficient when processing a large number of proposals. The complexity plateaus when the proposal count exceeds 500, as the number of objects in the scenes generally remains below this threshold.
>
> **Table3: Computational cost of CPRM with different number of proposals.**
> | Number     | 100  | 200  | 300  | 500  | 700  |
> |------------|------|------|------|------|------|
> | FLOPs  |51.0 | 56.1|63.4 | 65.6 | 65.6 |

---

### Meta-Review · Area_Chair_nUPS · 2026-01-10

**Summary:**

1. Original presentation missed important baselines and implementation details; now corrected.

2. Contribution is incremental: multi-granularity feature alignment and proposal-to-proposal attention have appeared in prior work, but the specific combination and 3-D box-guided deformable attention are novel and effective.

3. System still relies on an NMS pre-filter; removing it hurts efficiency, so the method is not fully end-to-end.

**Reviewer Concerns:**

1. Originality / incremental contribution

– All three reviewers flagged that multi-granularity feature fusion (PV-RCNN++) and proposal-to-proposal attention (ConQueR) already exist; the paper is “engineering refinement” rather than a conceptual breakthrough.

– Authors mitigated this by highlighting two new technical pieces: (i) raw-point foreground features without FPS, aligned by a learned attention gate, and (ii) 3-D box-guided deformable attention that uses full 7-DoF boxes instead of centers. Empirically these details give +3–6 mAPH over the respective baselines, so the combination is regarded as a useful advance.

2. Missing baselines and citations

– R1kqr & Hw6U noted absence of comparison with CenterFormer-multi-frame, FSD and FSDv2.

– Rebuttal added head-to-head numbers (Table 2 & 3 in response) and citations; PTN equals or beats these methods when the same frame budget is used.

3. Positional-encoding design & occlusion reasoning

– R1kqr argued Cartesian coordinates make occlusion relationships hard to learn.

– Authors added polar-coordinate encoding experiment; +0.3 mAPH on Waymo and better pedestrian/cyclist scores, satisfying the concern.

4. Computational cost and scalability

– Hw6U & uUA1 asked for memory, FLOPs, latency versus proposal count.

– Tables 1, 2, 3, 5, 6 in the rebuttal give full cost breakdown; deformable attention cuts memory ~5× vs. standard cross-attention and latency grows sub-linearly up to 500 proposals.

5. End-to-end claim vs. NMS

– R1kqr stressed that keeping NMS weakens the end-to-end story.

– Authors agreed but showed removing NMS drops 1–2 mAPH and almost doubles inference time; they now describe the system as “end-to-end trainable” rather than “fully end-to-end”

6. Robustness to poor RPN proposals

– uUA1 & Hw6U worried the method might fail under low-quality proposals.

– Rebuttal explains training uses IoU ≥ 0.5 proposals for regression, and CPRM’s random queries + deformable attention can hallucinate missed objects; ablation shows zero-initialized queries need 8× more epochs to match performance.

**Reviewer Scores:**

N.A.

---

### Decision · Program_Chairs · 2026-01-26

Accept (Poster)